# LEARNING FROM THE PAST WITH CASCADING ELIGIBILITY TRACES

Tokiniaina Raharison Ralambomihanta[*1,2], Ivan Anokhin[*1,3], Roman Pogodin[*1,4,5],
Samira Ebrahimi Kahou[†1,6,7], Jonathan Cornford[†1,8], Blake A. Richards[†1,4,5,7,9,10]

[1]Mila – Quebec Artificial Intelligence Institute
[2]Department of Bioengineering, McGill University
[3]Université de Montréal
[4]Department of Neurology & Neurosurgery, McGill University
[5]Montreal Neurological Institute, McGill University
[6]University of Calgary
[7]CIFAR AI Chair
[8]School of Computer Science, University of Leeds
[9]School of Computer Science, McGill University
[10]CIFAR Learning in Machines and Brains Program
[*]Equal contribution
[†]Co-senior authors
{toky.raharison-ralambomihanta, ivan.anokhin, roman.pogodin, ebrahims,
cornforj, blake.richards}@mila.quebec

## ABSTRACT

Animals often receive information about errors and rewards after significant delays. In some cases these delays are fixed aspects of neural processing or sensory feedback, for example, there is typically a delay of tens to hundreds of milliseconds between motor actions and visual feedback. The standard approach to handling delays in models of synaptic plasticity is to use eligibility traces. However, standard eligibility traces that decay exponentially mix together any events that happen during the delay, presenting a problem for any credit assignment signal that occurs with a significant delay. Here, we show that eligibility traces formed by a state-space model, inspired by a cascade of biochemical reactions, can provide a temporally precise memory for handling credit assignment at arbitrary delays. We demonstrate that these cascading eligibility traces (CETs) work for credit assignment at behavioral time-scales, ranging from seconds to minutes. As well, we can use CETs to handle extremely slow retrograde signals, as have been found in retrograde axonal signaling. These results demonstrate that CETs can provide an excellent basis for modeling synaptic plasticity.

## 1 INTRODUCTION

Learning requires a mechanism for assigning credit for errors and successes to past neural activity (Gerstner et al., 2018). In biological learning, the signals necessary for credit assignment in neural circuits arrive after a temporal delay, for instance via latency in sensory feedback following motor actions (Omrani et al., 2016; Scott, 2016). The source of delays can be generally categorized as either *external* or *internal*. External delays occur due to latency in environmental reward signals, for example receiving food or other external reinforcers, and as such can be variable. In contrast, internal delays arise from delays in the neural processing mediating the credit assignment calculation itself. Depending on the neural mechanisms involved, they are approximately fixed and can range from hundreds of milliseconds to several minutes (Fitzsimonds & Poo, 1998). Synaptic plasticity rules that model the credit assignment calculation, therefore, need to account for such fixed internal delays.

The traditional solution to internal delays in synaptic plasticity rules is to use exponentially decaying synaptic eligibility traces (ETs) (Gerstner et al., 2018; Shouval & Kirkwood, 2025), which are decaying records of synaptic activity. However, some experimentally observed synaptic plasticity

rules are tuned to fixed non-zero delays (Suvrathan et al., 2016; Shindou et al., 2019), and therefore do not fit with an exponential decay of credit eligibility. When the neural activity and the corresponding credit assignment signal are separated by few intervening events, such delays will have minimal impact on learning. However, in general, ongoing neural activity will override past activity relevant to the current reward or error signal. Therefore, traditional eligibility traces are not well-suited for the temporal scale of credit assignment delays in biological learning when the delay is non-zero and fixed.

To solve this problem we present a generalization of traditional eligibility traces. Inspired by synaptic biochemical cascades (Zhang et al., 2021), we model eligibility traces as state-space models that incorporate a cascade of synaptic memory traces. These cascading eligibility traces (CETs) provide a delayed and concentrated temporal window of maximal credit assignment. This model fits the experimentally observed unimodal delays, and is also consistent with biological mechanisms of synaptic plasticity (Friedrich et al., 2011; Fusi et al., 2005; Zhang et al., 2021).

We present a series of results that demonstrate the utility of CETs for credit assignment with biologically realistic delays. Specifically, we show that we can engage in both supervised and reinforcement learning in multi-layer networks under two distinct delay scenarios. First, we examine learning situations where the delays are consistent across layers of the network, as would be the case for various models of biological credit assignment in which a learning signal is broadcast across layers (e.g. direct feedback alignment (Nøkland, 2016) and other local learning rules (Frémaux & Gerstner, 2016; Ororbia, 2023)). Second, we show that CETs work when delays are stacked through the network, such that late layers receive credit signals sooner than early layers. We provide evidence that this approach works in both scenarios under a variety of biologically relevant delays, ranging from hundreds of milliseconds to minutes. Notably, the fact that CETs work when delays are stacked across layers and last for minutes shows that CETs could be applicable to credit assignment signals carried by retrograde axonal signals or neuropeptides, making this approach relevant for a number of biologically plausible credit assignment models (Liu et al., 2022; Fan & Mysore, 2024).

Altogether, our results indicate that CETs are a promising approach for handling delayed credit assignment signals in models of biological learning. More broadly, this provides a general framework for reasoning about synaptic memory in real neural networks.

## 2 RELATED WORK

Our work is related to and builds upon several strands of research on synaptic plasticity and biological credit assignment.

Eligibility traces (ETs) have long been a dominant framework for modeling how synaptic plasticity mechanisms may bridge temporal gaps between neural activity and feedback (Gerstner et al., 2018; Shouval & Kirkwood, 2025). Related theoretical extensions include ETs to approximate backpropagation through time (BPTT; Bellec et al. (2020)). Experimental evidence for ETs is well-established, with multiple studies reporting how synaptic changes can be induced by reward signals arriving seconds to minutes after neural activity (Brzosko et al., 2015; He et al., 2015; Bittner et al., 2017; Suvrathan, 2019).

In some experimental results, the timing of maximum synaptic change is tuned to specific delays (e.g. 120ms in the cerebellum (Suvrathan et al., 2016) and 2s in the striatum (Shindou et al., 2019)). This is in contrast to traditional ETs, and indicates a plasticity rule that encodes temporal structure in addition to the presence of past activity. One approach to model these findings is to combine two independent ETs for potentiation and depression to produce a composite ET that peaks at a required time delay (He et al., 2015; Huertas et al., 2016). This approach is conceptually similar to our CET model with 2 states. However, as we discuss below, this approach is restricted to producing ETs with a broad integration window, making it suitable for short delays only. As we show in our work, CETs with a larger number of states overcome this issue.

An example of extreme delays in plasticity-related signals is retrograde axonal signaling: i.e. "backward" propagation of chemical signal through the axon and synapses (Maday et al., 2014; Alger, 2002; Fitzsimonds & Poo, 1998). These signals play a role in activity-dependent synaptic plasticity at the level of individual synapses (Regehr et al., 2009), and have been suggested to coordinate plasticity across several neurons (Fitzsimonds et al., 1997; Hui-zhong et al., 2000; Du & Poo, 2004).

However, retrograde signals have generally been discarded as a component of credit assignment (e.g. Lillicrap et al. (2016)) because retrograde axonal signaling is extremely slow (on average $1.31\mu$m/s; Cui et al. (2007)), meaning that any error signal delivered via retrograde signaling would arrive minutes after the relevant neural activity. Nevertheless, there's been recent interest in this approach (Fan & Mysore, 2024). Here, we study to what extent delays on the order of retrograde process timescales could be compensated with CETs.

In parallel to neuroscience, the deep learning community has also worked on the the problem of delayed feedback from the perspective of decoupling the forward and backward passes for efficiency (Jaderberg et al., 2017; Malinowski et al., 2020). More generally, there is an extensive body of related work modeling how neural circuits may estimate and communicate credit in a biological plausible manner (Lillicrap et al., 2020). This includes credit computations with dendrites and bursts (Greedy et al., 2022; Payeur et al., 2021; Sacramento et al., 2017), and neuropeptides (Liu et al., 2022).

## 3 CASCADING ELIGIBILITY TRACES (CETs)

Synaptic plasticity for learning always requires some memory for presynaptic activity in the network. Consider a network containing a neuron with activity $z_t = f(\mathbf{x}_t^\top \mathbf{w})$, where $\mathbf{w}$ are the synaptic weights and $\mathbf{x}_t$ are the presynaptic inputs to the neuron at time $t$. To minimize a loss $L$, synaptic changes in $\mathbf{w}$ can follow the negative gradient over the loss $L$,

$$-\frac{\partial L(\mathbf{x}_t^\top \mathbf{w})}{\partial \mathbf{w}} = -\frac{\partial L(z_t)}{\partial z_t} f'(\mathbf{x}_t^\top \mathbf{w}) \mathbf{x}_t \equiv -\delta_t f'(\mathbf{x}_t^\top \mathbf{w}) \mathbf{x}_t \,. \tag{1}$$

This gradient-based formulation of plasticity Eq. (1) covers various forms of biological learning. For example, Hebbian learning can be recovered by using the loss $L(z_t) = -z_t^2$ and layer-wise learning rules can be defined similarly. And, of course, backpropagation follows the same chain rule logic for the loss defined over several layers of neurons.

Importantly, these updates require that the credit assignment signal $\delta_t$ is paired with the appropriate presynaptic inputs, $\mathbf{x}_t$. Hence, in the presence of any delays in computation of the credit assignment signal a learning system would face a temporal mismatch problem: if it takes $T$ seconds to calculate and propagate the credit assignment, then at time $t$ the error signal $\delta_t$ received by a neuron would have to be matched to an older presynaptic activity memory $\mathbf{x}_{t-T}$ (see Fig. 1A, top row). If learning is done in phases this need not be problematic. But, if learning happens online, as is likely the case in real brains, neural activity would correspond to the current time point, $\mathbf{x}_t$ only, so the previous presynaptic activity information, $\mathbf{x}_{t-T}$, would have to be somehow stored by the synapses.

Eligibility traces (ET) represent the classic solution to this problem: they add a memory component to the synapse that keeps track of recent activation for a single presynaptic neuron. Here, we will pick one weight $w^i$ and the corresponding input $x_t^i$, and discuss an ET $h_t^{\mathrm{ET}}$ such that changes in $w^i$ are proportional to $-\delta_t h_t^{\mathrm{ET}}$ (as in Eq. (1); dropping the index $i$ from $h_t^{\mathrm{ET}}$ for convenience).

Denoting the Hebbian-like term $h = f'(\mathbf{x}_t^\top \mathbf{w}) x_t^i$ (in the sense of it being a product of pre- and postsynaptic factors, $x_t^i$ and $f'(\mathbf{x}_t^\top \mathbf{w})$ correspondingly),

$$h_t^{\mathrm{ET}} = \int_0^t e^{-\gamma (t-s)} h_s \, ds \,, \tag{2}$$

where $\gamma > 0$ is a discount factor. The main advantage of ETs is that they're easy to implement:

$$\dot{h}_t^{\mathrm{ET}} = -\gamma h_t^{\mathrm{ET}} + h_t \,.$$

ETs effectively convolve the presynaptic activity $h_s$ with an exponential kernel $g(t)$, i.e.

$$h_t^{\mathrm{ET}} = (g * h)(t) = \int_0^t g(t - s) h_s \, ds \,, \tag{3}$$

and use this as a means of weighting past activity for combining it with credit assignment signals. One of the appeals of ETs as a solution to delayed credit assignment signals is that they do not require extensive memory, and are therefore a biologically plausible approach for learning.

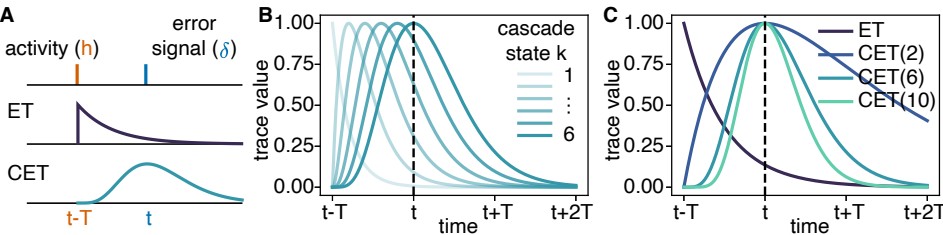

Figure 1: **A.** Learning with eligibility traces: neural activity $h$ is followed by a delayed error signal $\delta$. The standard eligibility trace (ET) is an exponentially decaying trace of $h$ that can be matched to the error signal at time $t$. The cascading ET (CET) reflects $h$ like a regular ET, but peaks at the required time $t$. **B.** Time evolution of each state of a 6-state CET with a delay $T$ and a unit input at $t - T$. **C.** Comparison of a standard ET and CETs with 2/6/10 states representing delay $T$ for a unit input at $t - T$.

Notably, the classic form of ET assigns the maximal trace values to the most recent time-points $s = t$ in Eq. (3). That is appropriate for situations in which there are few intervening presynaptic events between times $t$, when the credit assignment signal arrives, and $t - T$, when the presynaptic activity occurred (as in Fig. 1A). But, if $h_t$ changes frequently relative to the delay in the credit assignment signal then gradients calculated with classic ETs, i.e. $\delta_t\, h_t^{\mathrm{ET}}$, can be a poor approximation of the true gradient, $\delta_t\, h_{t-T}$.

Ideally, when we consider the long delays faced by biological learning agents we would have ETs satisfying two conditions. First, the maximal value of the synaptic trace should occur at a delay of $s = t - T$, rather than $s = t$ in Eq. (3). Second, it is better if we can use a more precise weighting of the past, i.e. if the ET values $g(t - s)$ are as small as possible for anything other than $s = t - T$.

Classic ETs do not provide these characteristics. Combined LTP & LTD eligibility traces (He et al., 2015; Huertas et al., 2016) satisfy the first condition, but not the second. They effectively take a difference of two standard ETs in Eq. (2) to convolve past activity with $g(t) = \exp(-\gamma_\mathrm{P} t) - \exp(-\gamma_\mathrm{D} t)$ (as in Eq. (3)). While this $g$ peaks with a delay $T = (\log \gamma_\mathrm{D} - \log \gamma_\mathrm{P})/(\gamma_\mathrm{D} - \gamma_\mathrm{P})$, it keeps a large weight for more recent points. Additionally, we show in Section A.3 that the limiting sequence of $\gamma_\mathrm{D}, \gamma_\mathrm{D}$ for minimizing the second moment around $T$ constrained to a maximum at $T$ converges to our proposed method.

As a flexible solution to both of these problems, we propose using ETs constructed from a simple state-space model:

$$\dot{h}_t^1 = -\alpha\, h_t^1 + h_t\,,$$
$$\cdots$$
$$\dot{h}_t^k = -\alpha\, h_t^k + h_t^{k-1}\,, \qquad (4)$$
$$\cdots$$
$$\dot{h}_t^{\mathrm{CET}} = -\alpha\, h_t^{\mathrm{CET}} + h_t^{n-1}\,.$$

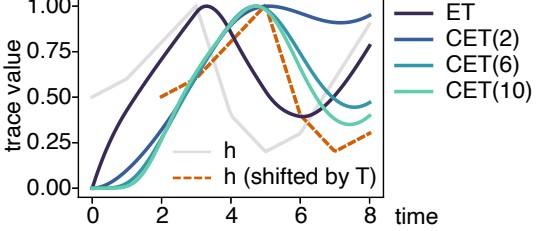

Figure 2: Representation of the input signal $h$ (gray) with a $T = 2$ delay (dashed orange line) using a standard eligibility trace (ET) and cascading ETs (CETs) of different orders. Only CETs of higher order reflect the time evolution of the input (i.e. match the orange line).

Eq. (4) can be used as a model of a cascade of biochemical reactions. This could involve, for example, a cascade of phosphorylation processes or enzymatic reactions (Zhang et al., 2021). Implementing CET($n$) online requires $n$ states per synapse each with a constant update cost so the computational costs scale linearly in time and memory.

This form of CET gives us the following formulation (see Section A for a derivation):

$$h_t^{\mathrm{CET}} = \frac{1}{(n-1)!} \int_0^t (t - s)^{n-1} e^{-\alpha\,(t - s)}\, h_s\, ds\,, \qquad (5)$$

which for $\alpha = \frac{n-1}{T}$ convolves the presynaptic activity with a kernel $g(t) \propto t^{n-1} e^{-\alpha t}$ that peaks at $t = T$ (Fig. 1C). For classical ETs (which correspond to a single-state model of $n = 1$) we either set the decay to be $\alpha = \frac{1}{T}$ (for supervised learning) or we conduct a grid search on this hyperparameter (for reinforcement learning).

The dynamical system in Eq. (4) is defined by two parameters: the number of states $n$ and the decay term $\alpha$. Increasing $n$ while keeping the peak-time fixed leads to a narrower kernel (see Fig. 1C for a visualization of different kernels $g(t)$), but having even two states (instead of one for standard ETs) can account for delayed signals. However, to accurately represent delayed signals, more states are typically needed (see Fig. 2 for time domain response and Section A.2 for Laplace domain analysis).

## 4 EXPERIMENTS

We illustrate the influence of CETs on learning with delays in two scenarios[1]: (1) learning with delays on behaviorally relevant timescales (e.g. on the order of seconds) in Section 4.1; (2) credit propagation through very slow chemical signals (e.g. retrograde axonal signaling (Fitzsimonds et al., 1997)) in Section 4.2. In our simulations we assume that each input lasts for 200 ms, which is roughly one saccade or one theta cycle in the brain (Young & Stark, 1963). Thus, a single time-step in the simulation is treated as a 200 ms, so a delay of $T = 1$ s would mean that the $\delta$ signal arrives 5 time-steps after the input is initially presented to the network. Put another way, with a simulated delay of $T = 1$ s there are 4 image presentations that occur after the initial image presentation and before the $\delta$ signal for that image arrives.

In (1), we assume that the error signal $\delta$ is propagated to all layers simultaneously since credit signals propagated via action potentials could be transmitted to the entire network in parallel. As well, we calculate the error signal explicitly, but we note that this calculation could easily be substituted with any of the available mechanisms for biologically plausible error calculation (e.g. 3-factor Hebbian learning rules Frémaux & Gerstner (2016)).

In (2), we consider a much longer delay $T$ of 2 minutes, which corresponds to roughly the amount of time it takes for chemical signals to travel backwards along the axon. At the speed of $1.31 \, \mu$m/s (Cui et al., 2007), this covers roughly $\sim 160 \, \mu$m, corresponding to the typical $< 200 \, \mu$m distance in the cortex (Song et al., 2005; Cui et al., 2007). As well, in-line with propagation of a retrograde signal, we assume that the delays stack up over layers. Thus, the last layer has no delay, the penultimate layer has a delay of $T = 2$ minutes, the next layer has a delay of $T = 4$ minutes, and so on. Thus, each preceding layer's delay is increased by $T$.

We use two types of tasks: supervised image recognition on MNIST (LeCun, 1998) and CIFAR-10 (Krizhevsky et al., 2014), and reinforcement learning on state-based environments (namely CartPole and LunarLander), as well as on a more complex visual environment (namely MinAtar/SpaceInvaders (Young & Tian, 2019), which use raw pixel observations as input). We use a 3-layer MLP (*input* $\rightarrow 512 \rightarrow 512 \rightarrow 10$) for MNIST and a small CNN with 3 convolutional layers (*input* $\rightarrow 32 \rightarrow 64 \rightarrow 128$) and two linear layers ($512 \rightarrow 10$) for CIFAR-10. For RL, we use a 3-layer MLP with a hidden dimension of 256 that we train with the Actor-Critic method, and report results over 3 seeds. To simplify training in the delayed setup, only the Actor is trained with a delayed error signal, while the Critic is updated via standard backpropagation. The Actor is trained using an online implementation of the $\lambda$-return via *RL eligibility traces* (Sutton & Barto, 2018). Other experimental details (hyperparameters, compute resources) can be found in Section C. The PyTorch (Paszke et al., 2019) implementation and experiments are provided in the Supplementary Material.

### 4.1 LEARNING WITH DELAYS ON BEHAVIORALLY RELEVANT TIMESCALES

On MNIST, we observed that classical ETs (corresponding to a CET with one state) maintain strong performance up to delays of two seconds, i.e. up to 10 image presentations before a $\delta$ arrives (Fig. 3, left). This shows that classical ETs can remain effective for relatively simple tasks and short delays. However, their performance breaks down at longer delays of $T \geq 4$ s. At these longer delays we can see that increasing the number of states in the CETs improves performance, and can keep the accuracy level high at up to 10 s delay (50 image presentations). Past this point, we found that only

---

[1]Code is available at: `https://github.com/avecplezir/CET`

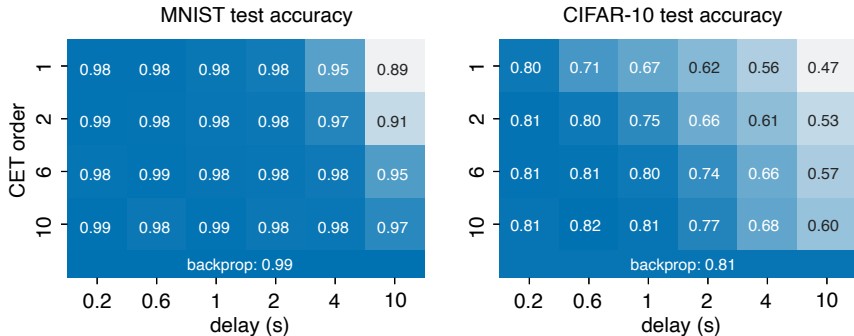

Figure 3: Accuracy for MNIST and CIFAR-10 datasets across varying numbers of CET states and delays on behaviorally relevant timescales. A single state (top row) corresponds to standard ETs.

a perfect eligibility trace (i.e. an infinite number of states corresponding to a Dirac delta memory) would preserve performance.

The results with CIFAR-10 were even more pronounced (Fig. 3, right). Classical ET performance deteriorates at any delay tested and rapidly decreases, showing that more complex visual tasks are less robust to imprecise time resolution. A key observation is the gradient in performance, with accuracy generally improving with the number of states in the CETs and decreasing with delays. This trend reflects how the CET impulse response becomes increasingly concentrated around the target delay as the number of states increases, which provides finer temporal resolution. The same trends also holds on a more challenging dataset: see Section F in the Appendix for TinyImageNet performance.

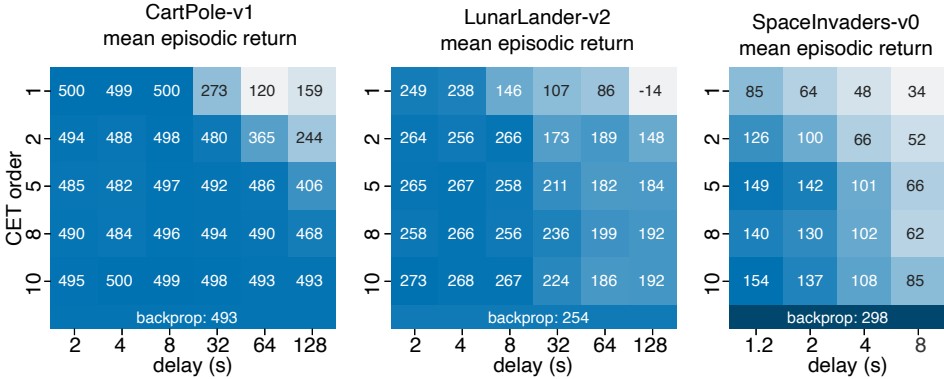

Figure 4: Mean episodic return for different RL environments across varying numbers of CET states and delays on behaviorally relevant timescales. A single state (top row) corresponds to standard ETs.

We observe the same trend for the RL tasks (see Fig. 4): shorter delays and a higher number of CET states result in better performance. Note that CartPole and LunarLander are simple RL tasks and remain solvable even with long delays and fewer states in the CETs. In contrast, MinAtar/SpaceInvaders (Fig. 4, right) is a more complex, image-based environment where performance begins to degrade more quickly when a delay is introduced. In fact, CET performance is at best only half the performance of a perfect memory, even at the shortest delay we tested. We therefore hypothesize that precise credit assignment, without mixing nearby time points, is especially important for complex, non-i.i.d. tasks. Altogether, our results demonstrate that at behaviorally relevant time delays higher-order CETs can greatly enhance performance beyond that achieved by classical ETs, particularly at long delays. However, they cannot fully compensate for delays in highly complex, non-i.i.d. tasks.

To better understand the reasons for the performance we examined how well the weight updates were aligned to the true gradient. We measured the cosine similarity, $\mathbf{a}^\top \mathbf{b}/(\|\mathbf{a}\| \|\mathbf{b}\|)$, between the vector of weight updates given by our CETs, $\mathbf{a}$, and the true gradient as calculated by backpropagation, $\mathbf{b}$. In Fig. 5, we plot cosine similarity for all CET models for delays of 1, 2, and 4 s on CIFAR-10

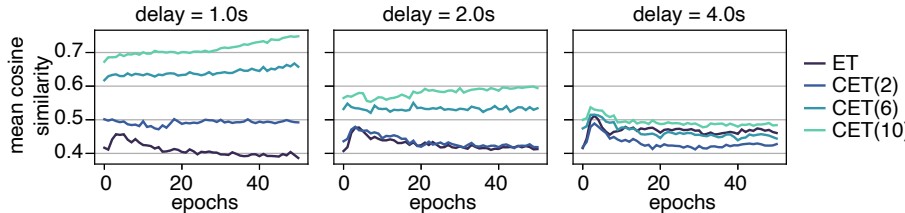

Figure 5: Average cosine similarity over all layers between true gradients and gradients computed with either ETs or CETs for the CIFAR-10 dataset.

(as performance differences were noticeable across these delays in Fig. 3). We observed that at all times during training, and at shorter delays, an increase in the number of states in the CETs lead to better alignment with the true gradient (Fig. 5, left and center). However, as the delay increases, the alignment drops even for higher-order CETs (e.g. with 10 states; Fig. 5, right). When we broke this down by layer, we observed similar patterns (Section D).

Finally, we studied if CETs can handle variable and unknown delays. First, we compared the performance of CETs and ETs in situations where reward delay followed uni-modal distributions with differing variance. We found that CETs always outperformed ETs (Section L in the Appendix), though the benefit decreased as the variance increased and the delay distribution became closer to uniform. This is expected since the kernel shape should me matched to the delay distribution. Indeed, the standard ET can approximate the uniform delay well enough so the advantage of CETs is reduced. Second, for unknown delays, we found that $\alpha$ (and hence the position of the CET's peak) can be learned using weight perturbation, outperforming ETs (see Section L.4 in the Appendix).

## 4.2 COMPUTATION WITH EXTREMELY LONG DELAYS FOR RETROGRADE AXONAL SIGNALING

We next investigated the possibility of using CETs to model situations with very long, and accumulating, delays. Here, the goal was to consider delays introduced by chemical signals (e.g. retrograde axonal signaling) which could in principle be used for credit assignment (Fan & Mysore, 2024), but would take minutes to propagate from synapses back to cell bodies. We assume that we do not have to solve the weight transport problem of backpropagation (Grossberg, 1987), since retrograde signals could easily have access to synaptic weight values (Fan & Mysore, 2024). (See Discussion.) However, the approach with CETs here could also easily be used in conjunction with other solutions to weight transport, including feedback alignment (Lillicrap et al., 2016) or feedback learning mechanisms (Akrout et al., 2019). Additionally, we assume that the calculation of the $\delta$ signals has access to the post-synaptic activation derivative $f'(\mathbf{x}^\top \mathbf{w})$ at the appropriate delay, which implies another memory mechanism at the soma, rather than the synapse. This could be modeled with CETs as well, but we leave that for future work.

An additional consideration that we took into account here is that if credit signals were propagating backwards via retroaxonal biochemical transmission, then error signals would take progressively more time as they travel across a number of synaptic steps (i.e. network depth). Therefore, in a feedforward network if we assume that a single layer takes $T$ time to propagate the error signal backwards, then a layer $m$ synaptic steps back will receive the error signal at time $t = (m-1)T$ (Fig. 6).

Finally, to handle very long delays on visual tasks (MNIST, CIFAR-10), we make an additional assumption: *the CETs are modulated by an additional "salience signal" that zeros out the input to the CETs unless the loss is very large*. This mechanism reduces the number of inputs being stored in CETs, in line with work on reducing the energetic costs of plasticity (van Rossum & Pache, 2024). This assumption is also consistent with theories of predictive processing and surprise-driven learning, in which events with high prediction error drive plasticity (Friston, 2005;

Figure 6: Backprop using retrograde axonal signaling results in delay accumulation: in the last layer, the forward and backward signal are computed simultaneously at time $t$. Each consecutive layer of the backwards pass takes $T$ more seconds (time taken by retrograde signaling).

Itti & Baldi, 2009; Dabney et al., 2020). We use 1.25 % of points with the largest losses in a batch (keeping their positions in the batch, such that position index encodes time; see Section B.2 for details). To handle long retrograde signal delays in RL without sparsification, we simplify the setting by assuming each time step lasts 300 ms instead of 200 ms as in other our experiments. This corresponds to a delay of 400 frames for the second layer and 800 frames for the first layer.

When training networks on visual tasks with large stacking delays across layers, we observed that the performance increased with increasing CET order (Fig. 7). Moreover, networks with different CET orders trained at markedly different rates, with higher orders learning faster (Fig. 7). This was more pronounced for the deeper convolution network trained on CIFAR-10 (Fig. 7B) than the shallow MLP trained on MNIST (Fig. 7A). Given that the last layer was trained without delay in these experiments, these results must be due to the impact of delays on learning in the intermediate representations.

To understand the performance differences for different orders of CET we analyzed gradient alignment during training. Because networks learned at different speeds, which affects the dynamics of gradient alignment (Section D), we used test accuracy as the independent variable, rather than training iteration. On both MNIST and CIFAR-10 we observed that gradient alignment degrades for the earlier layers, as is expected for the increasing delays (Fig. 7C,D). Across all layers, increasing CET order was associated with an increase in gradient alignment, in line with task performance. However, higher orders were unable to fully recover alignment with the gradient.

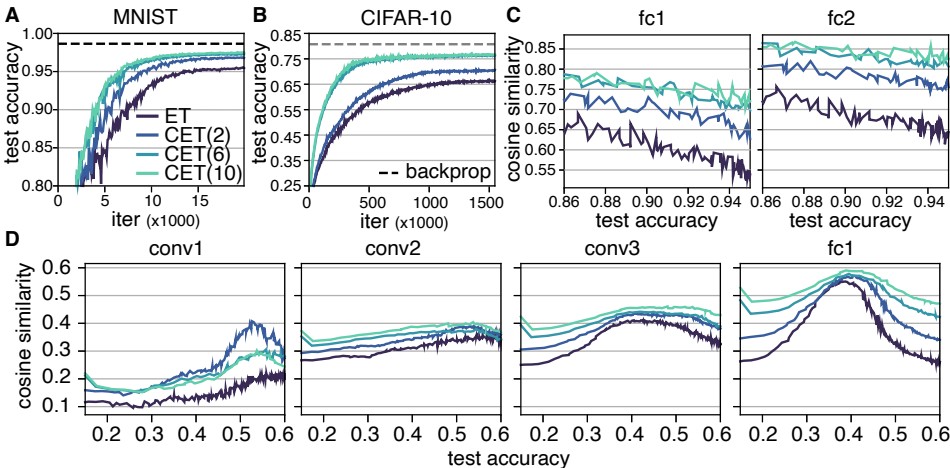

Figure 7: **A.** Test accuracy on MNIST as a function of number of CET states for the retrograde experiments. **B.** Same as **A**, but on CIFAR-10. **C.** Cosine similarity between the true gradient and the ET/CET approximation across different test accuracies (see **A**). Each plot shows an individual layer of an MLP during training on MNIST. **D.** Same as **C**, but each plot shows an individual layer of a CNN during training on CIFAR-10.

For the RL tasks, we observed similar trends. Increasing the number of CET states leads to improved performance on both CartPole and LunarLander (Fig. 8A-B). As before, the increase in the number of CET states also led to increased alignment with the true gradient (Fig. 8C-D), although mostly in the second layer, which helps to explain the improved performance.

Altogether, our results demonstrate that when delays in credit signals are very long (on the order of minutes), and stacked (summing for each synaptic step), CETs can be used to store memory for previous activity in order to accurately estimate gradients and learn. As such, CETs would, in principle, permit credit assignment in situations where errors are propagated backwards via very slow chemical retrograde signals (Fitzsimonds et al., 1997). However, there is a depth limit beyond which the delay would be too large to accurately approximate the gradient.

## 5 DISCUSSION

For organisms to learn, their brains must have mechanisms for handling delays between learning signals and past neural activity. Here we presented cascading eligibility traces (CETs), a generalization

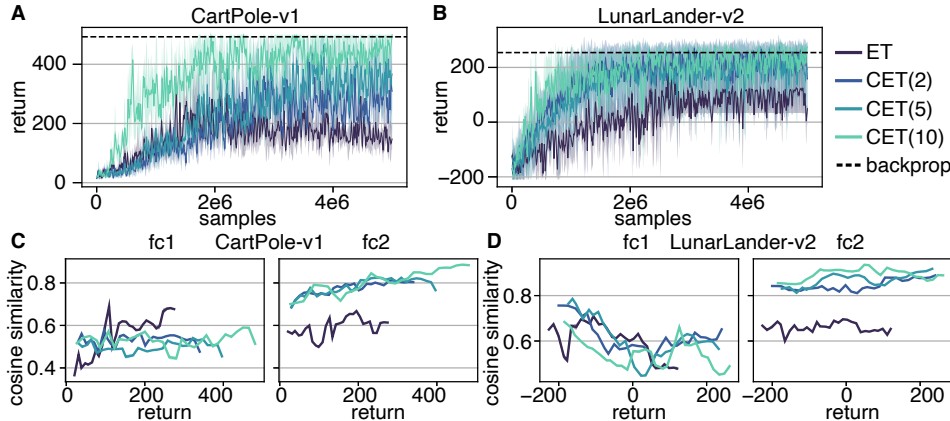

Figure 8: **A.** Episodic return on CartPole-v1 during training in the retrograde experiments. Solid lines: mean (3 seeds); shaded area: min/max values; dashed line: mean final backprop performance. **B.** Same as **A** but for LunarLander-v2. **C.** Mean cosine similarity (over cosine similarity values assigned to binned return values) between the true gradient and the ET/CET approximation w.r.t. return values in **A.**. **D.** Same as **C**, but for LunarLander-v2.

of classical eligibility traces and an abstract model of interacting biochemical processes within cells, as a candidate mechanism for bridging such delays. We showed that CETs enable learning over long delays on standard image datasets and RL settings, and found that by increasing the number of states in the eligibility trace cascade (with 1 state being equivalent to classical ETs) learning performance can be maintained with delays on the order of seconds. We explored how CETs contribute to the ongoing discussion around biologically plausible implementations of backpropagation (Fan & Mysore, 2024; Liu et al., 2022). Here we tested the hypothesis that synaptic CETs and learning from salient examples enable slow cytoskeletal retroaxonal signals to carry gradient information recursively over layers – an idea popular over two decades ago but since discarded (Harris, 2008). Again, we found CETs with a larger number of states improved performance. Though accumulating delays with network depth was still problematic for learning, CETs demonstrate that learning over timescales relevant for slow chemical signaling is feasible. Our experiments with CETs validate retroaxonal signals a potential solution to the credit assignment problem.

Our work was limited to experiments with feedforward architectures. While extending it to more biologically relevant scenarios, such as RNNs or spiking networks, is beyond the scope of this work, preliminary experiments with recurrent reservoir networks (Appendix E) and leaky intergate-and-fire neurons (Appendix J) show promising results.

Classical ETs are exponentially decaying "memories" of synaptic activation that are thought to be implemented by activation of biochemical processes, such as CaMKII activation or other protein kinases (such as PKA, PKC, ERK, MAPK), which are typically triggered by the activation of G-protein coupled receptors (Gerstner et al., 2018). While the complexity of such interacting pathways has been recognized, there has been very little work exploring interactions between such biochemical processes for learning (though see (Friedrich et al., 2011; Huertas et al., 2016)). In this context, we are building off work exploring how complex interactions between kinase cascades mediates plasticity (Zhang et al., 2021). More generally, CETs provide a normative explanation for the complexity of cascade interactions in the context of learning—the improved performance with higher order CETs could explain why cells use biochemical cascades rather than a single biochemical signal.

One of the well-known biological implausibilities of backpropagation is that it requires that weights in the forward pass be reused in the backward pass. In the context of biology, this algorithmic requirement is known as the weight transport problem (Grossberg, 1987). Retroaxonal signals provide a potential solution to this problem because they pass back through the very same synapses used in the forward pass. As such, they could, in principle, carry information about the synaptic weights, thereby solving the weight transport problem (Fan & Mysore, 2024). However, the challenge with retroaxonal signals is that they are very slow, taking minutes to pass from the synapse to the cell body (Fitzsimonds & Poo, 1998). As we showed here, CETs provide a potential mechanism for

making learning at such delays feasible. Therefore, they open up the possibility of using retroaxonal signals for credit assignment. However, our results also showed that you can only stack such long delays over a few synaptic steps before learning deteriorates significantly, which would suggest that if retroaxonal signals are used for learning in the brain they would only be used for learning at relatively shallow "depths". Indeed this is consistent with experimental findings: for example Hui-zhong et al. (2000); Fitzsimonds et al. (1997) only found retroaxonal potentiation and depression over one "layer" in cultured neurons (i.e. one synaptic step). Although it was originally suggested that the lack of further propagation may be due to the size of the plasticity change (Fitzsimonds et al., 1997), our results provide evidence that recursive propagation delays are problematic, even with CETs, and would require additional mechanisms such as direct reward signaling (Nøkland, 2016).

For the retrograde experiments, we also relied on a salience signal to gate which presynaptic events were stored in the CETs. Conceptually, this mechanism reflects a broader class of theories in neuroscience and machine learning in which a subset of events—typically those associated with large prediction errors—drive plasticity. Three-factor learning rules, for instance, can encode surprise, effectively shutting down learning when no unexpected outcome occurs (Gerstner et al., 2018). Similarly, normative accounts link large prediction errors to high information gain (Friston, 2005; Itti & Baldi, 2009). Our use of top-k losses operationalizes this principle by ensuring that only high-salience presynaptic events are retained in the CETs. While we do not commit to a specific biological mechanism, this abstraction is consistent with proposals in which global neuromodulators signal behavioral relevance (Dabney et al., 2020) or dendritic compartments represent mismatches between predictions and inputs (Sacramento et al., 2017; Aizenbud et al., 2025).

In summary, our work on CETs provides an extension to the classical ET approach for handling delays between activity and feedback error signals or rewards. We have demonstrated that cascades of biochemical processes could be used by cells to store more temporally precise memories of past cell activity. These memories could then be combined with delayed error signals to estimate loss gradients. Therefore, our work provides another potential means of understanding how the brain can learn complicated tasks in a biologically plausible manner.

## ACKNOWLEDGMENTS AND DISCLOSURE OF FUNDING

This work was supported by: NSERC (Discovery Grant: RGPIN-2020-05105; Discovery Accelerator Supplement: RGPAS-2020-00031; Arthur B. McDonald Fellowship: 566355-2022); CIFAR (Canada AI Chair; Learning in Machine and Brains Fellowship); AccelNet (International network for brain-inspired computation, NSF 2019976); Canada Excellence Research Chairs (CERC) Program. This research was enabled in part by support provided by (Calcul Québec) (`https://www.calculquebec.ca/en/`) and the Digital Research Alliance of Canada (`https://alliancecan.ca/en`). The authors acknowledge the material support of NVIDIA in the form of computational resources. The research was enabled in part by computational resources provided by Mila - Quebec Artificial Intelligence Institute.

IA thanks Stephen Chung for his many helpful discussions and valuable comments on the reinforcement learning components of the project.

**Reproducibility Statement.** Code is available at: `https://github.com/avecplezir/CET`, with instructions for running the main experiments. Appendix C describes details for all hyperparameter configurations, model architectures, and the training pipeline.

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

## A    CASCADING ELIGIBILITY TRACES DERIVATIONS

### A.1    UPDATE DERIVATION

The ET we presented in Eq. (4) has the form

$$\dot{\mathbf{x}}(t) = \mathbf{A}\,\mathbf{x}(t) + \mathbf{b}(t)\,,$$

which are solved by

$$\mathbf{x}(t) = \exp(\mathbf{A}t)\,\mathbf{x}(0) + \int_0^t \exp(\mathbf{A}(t-s))\,\mathbf{b}(s)\,ds\,. \tag{6}$$

While the matrix exponent $\exp(\mathbf{A}t)$ is hard to compute in general, in our case,

$$\begin{pmatrix} \dot{x}_t^1 \\ \cdots \\ \dot{x}_t^{n-1} \\ \dot{x}_t^{\text{CET}} \end{pmatrix} = \begin{pmatrix} -\alpha & 0 & 0 & \cdots & 0 \\ \cdots & \cdots & \cdots & \cdots & \cdots \\ 0 & \cdots & 1 & -\alpha & 0 \\ 0 & \cdots & 0 & 1 & -\alpha \end{pmatrix} \begin{pmatrix} x_t^1 \\ \cdots \\ x_t^{n-1} \\ x_t^{\text{CET}} \end{pmatrix} + \begin{pmatrix} x_t \\ \cdots \\ 0 \\ 0 \end{pmatrix}\,, \tag{7}$$

therefore $\mathbf{A} = \alpha\,\mathbf{I}_n + \mathbf{N}$ for a nilpotent $\mathbf{N}$ (i.e. $\mathbf{N}^n = 0$), hence

$$\exp((\alpha\,\mathbf{I}_n + \mathbf{N})\,t) = \exp(\alpha\,t)\left(\sum_{i=0}^{n-1} \frac{1}{i!} N^i t^i\right)$$

$$= \exp(\alpha\,t) \begin{bmatrix} 1 & 0 & \cdots & 0 & 0 \\ t & 1 & 0 & \cdots & 0 \\ \frac{t^2}{2!} & t & 1 & \cdots & 0 \\ \cdots & & & & \\ \frac{t^{n-1}}{(n-1)!} & \frac{t^{n-2}}{(n-2)!} & \cdots & t & 1 \end{bmatrix}\,.$$

Therefore, for $\mathbf{x}(0) = 0$ and $\mathbf{b}(t)$ being non-zero only for the first coordinate, the last coordinate of $\mathbf{x}$ implements

$$x_n(t) = \int_0^t \exp(\alpha\,(t-s))\frac{(t-s)^{n-1}}{(n-1)!}\,b_0(s)\,ds\,.$$

Moreover, if $\mathbf{b}(t)$ is a step-wise function taking on a new value every $\Delta t$ points, a single step of the integration between $t$ and $t + \Delta t$ can be computed (exactly) using Eq. (6) as

$$\mathbf{x}(t + \Delta t) = \exp(\mathbf{A}\Delta t)\,\mathbf{x}(t) + \left[\int_0^{\Delta t} \exp(\mathbf{A}(\Delta t - s))\,ds\right]\mathbf{b}(t)$$

$$= \exp(\mathbf{A}\Delta t)\,\mathbf{x}(t) + \left[\int_0^{\Delta t} \exp(\mathbf{A}\,s)\,ds\right]\mathbf{b}(t)\,.$$

As $\mathbf{A}$ is non-singular, we can integrate this solution further to obtain

$$\mathbf{x}(t + \Delta t) = \exp(\mathbf{A}\Delta t)\,\mathbf{x}(t) + [\exp(\mathbf{A}\,\Delta t) - \mathbf{I}]\,\mathbf{A}^{-1}\mathbf{b}(t)\,. \tag{8}$$

### A.2    LAPLACE DOMAIN ANALYSIS

To clarify the effect of increasing $n$, we include a brief Laplace domain analysis. We can show that the Laplace transform converges pointwise to the Laplace transform of the perfect delay. Therefore, the outputs $u * h_n$ converge to $u(t - T)$ for finite-bandwidth low-frequency signals since the system acts as a low-pass filter, which explains why increasing $n$ sharpens the kernel and yields better results.

**Proposition 1.** *Let*

$$h_n(t) = Z_n^{-1} t^n e^{-nt/T}, \qquad Z_n = \int_0^\infty t^n e^{-nt/T}\,dt.$$

*Then the Laplace transform $\mathcal{L}\{h_n\}(s)$ converges pointwise to $e^{-sT}$ as $n \to \infty$.*

*Proof.* We first compute the normalization constant:

$$Z_n = \int_0^\infty t^n e^{-nt/T} \, dt = \frac{\Gamma(n+1)}{(n/T)^{n+1}}.$$

The Laplace transform of $h_n$ is

$$\mathcal{L}\{h_n(t)\}(s) = Z_n^{-1} \int_0^\infty t^n e^{-nt/T} e^{-st} \, dt.$$

Evaluating the integral gives

$$\mathcal{L}\{h_n\}(s) = \frac{(n/T)^{n+1}}{\Gamma(n+1)} \frac{\Gamma(n+1)}{(s+n/T)^{n+1}} = \left(\frac{n/T}{s+n/T}\right)^{n+1} = \left(1 + \frac{sT}{n}\right)^{-(n+1)}.$$

Taking the limit,

$$\lim_{n\to\infty} \mathcal{L}\{h_n\}(s) = \lim_{n\to\infty} \left(1 + \frac{sT}{n}\right)^{-(n+1)} = e^{-sT},$$

which is the Laplace transform of the ideal delay $\delta(t - T)$. □

## A.3 CONNECTION TO COMBINED LTP & LTD TRACES

**Proposition 2.** *The $L_1$ normalized LTP-LTD kernel over $[0, \infty)$ is given by*

$$h(t) = \frac{ab}{b-a} \left(e^{-at} - e^{-bt}\right),$$

*and we wish to find $a, b$ to concentrate the mass around $T$. When viewing $h(t)$ as a probability density function we want to minimize $f(a,b) = \mathbb{E}_h\left[(t - T)^2\right]$*

$$f(a,b) = \frac{1}{a^2} + \frac{1}{b^2} + \left(\frac{1}{a} + \frac{1}{b} - T\right)^2$$

*subject to $h$ reaching a maximum at $T$ or equivalently with the constraint*

$$0 = g(a,b) = \begin{cases} \dfrac{\ln(a) - \ln(b)}{a - b} - T, & a \neq b, \\ \dfrac{1}{a} - T, & a = b. \end{cases}$$

*Then the infimum over $\{(a,b) : g(a,b) = 0, \ a \neq b, a > 0, b > 0\}$ is achieved only in the limit $a, b \to 1/T$, with such a sequence converging pointwise and in $L_1$ to $\frac{1}{T^2} t e^{-t/T}$.*

*Proof.* Solving the constraint $g(a,b) = 0$ for $(a,b)$ in terms of a single real parameter $x$ yields the smooth parametrization

$$a(x) = \frac{x e^{Tx}}{e^{Tx} - 1}, \qquad b(x) = \frac{x}{e^{Tx} - 1},$$

valid for $x \neq 0$, with the continuous extension $a(0) = b(0) = 1/T$.

Substituting this parametrization into the objective gives

$$f(x) = f(a(x), b(x)) = \frac{(1 - e^{Tx})^2}{x^2} + \frac{(1 - e^{Tx})^2 e^{-2Tx}}{x^2} + \frac{\left(-Txe^{Tx} + e^{2Tx} - 1\right)^2 e^{-2Tx}}{x^2}.$$

We can directly check that $f(x)$ is an even function, hence $x = 0$ is an extreme point.

A computational verification shows that the extended function is strictly convex and therefore attains its unique minimum at $x = 0$.

It follows that the objective is minimized by taking a sequence $(a_n, b_n)$ with

$$a_n, b_n \to \frac{1}{T}, \qquad a_n \neq b_n.$$

It remains to show that such a sequence converges to $\frac{1}{T^2} t e^{-t/T}$.

Let $f(\lambda, t) = e^{-\lambda t}$. By the mean value theorem, there exists $c \in (a_n, b_n)$ such that

$$\frac{\partial}{\partial \lambda} f(\lambda, t)\Big|_c = -t e^{-ct} = \frac{e^{-at} - e^{-bt}}{a - b}$$

$$t e^{-ct} = \frac{e^{-at} - e^{-bt}}{b - a}.$$

Then for any sequence $(a_n, b_n)$ with $|a_n - b_n| \to 0$ and $a_n \to 1/T$, the corresponding kernels $h_n$ converge pointwise to the desired limit. Since $0 \leq h_n(t) \leq C e^{-t/(T+1)^2}$ for all sufficiently large $n$, dominated convergence ensures $L_1$ convergence. □

## B  IMPLEMENTATION DETAILS

### B.1  ALTERNATIVE EXPRESSION FOR STEPWISE INPUTS

For experiments on visual tasks, we consider that a batch of inputs corresponds to a time-series where the batch dimension corresponds to the time dimension. In this case, the output of the state-space model in Eq. (4) can be obtained via a discrete convolution, which can be efficiently computed as a matrix multiplication.

Starting from Eq. (8), we can denote $\mathbf{M} = \exp(\mathbf{A}\Delta t)$ and $\mathbf{K} = [\exp(\mathbf{A}\,\Delta t) - \mathbf{I}]\,\mathbf{A}^{-1}$, such that

$$\mathbf{x}(t + \Delta t) = \mathbf{M}\,\mathbf{x}(t) + \mathbf{K}\,\mathbf{b}(t),$$

and therefore

$$\mathbf{x}(k\,\Delta t) = \mathbf{M}^k\,\mathbf{x}(0) + \sum_{i=0}^{k} \mathbf{M}^{k-i}\mathbf{K}\,\mathbf{b}(i\,\Delta t).$$

If we assume the initial state was zero, $\mathbf{x}(0) = 0$, the SSM outputs $x_{k\,\Delta t}^{\text{CET}}$ will be computed as (dropping the SSM superscript for convenience)

$$
\begin{bmatrix} x_0 \\ x_{\Delta t} \\ \vdots \\ x_{k\,\Delta t} \end{bmatrix} =
\begin{bmatrix} g_0 & 0 & \cdots & 0 \\ g_1 & g_0 & \cdots & 0 \\ \cdots & & & \\ g_k & g_{k-1} & \cdots & g_0 \end{bmatrix}
\begin{bmatrix} b_0 \\ b_{\Delta t} \\ \vdots \\ b_{k\,\Delta t} \end{bmatrix} = \mathbf{G}\mathbf{b}[t]
\tag{9}
$$

where $g_j = (\mathbf{M}^j\mathbf{K})_{n0}$.

Alternatively, to get a closed form expression for $g$, we may rewrite the stepwise constant input as the convolution of the appropriate impulse train with the rectangular function. Using $\delta$ for the Dirac delta and $\theta$ for the Heaviside function, we have

$$b(t) = \sum_{i=1}^{\infty} a_i \delta(t - t_i), \ \text{rect}(t) = \theta(t) - \theta(t-1)$$

$$\hat{b}(t) = \sum_{i=1}^{\infty} a_i \text{rect}(t - t_i) = (b * \text{rect})(t)$$

$$x(t) = \hat{b} * g = b * (\text{rect} * g)$$

$$= \sum_{i=1}^{\infty} a_i (\text{rect} * g)(t - t_i).$$

We can therefore compute the exact continuous-time output with a discrete convolution using $\hat{g} = \text{rect} * g$.

$$g(t) = \theta(t)kt^n e^{-\frac{n}{T}t}$$

$$\text{rect} * g = \int_{-\infty}^{\infty} h(\tau)\text{rect}(t - \tau)d\tau$$

$$= k \int_{\max(0,t-1)}^{t} \tau^n e^{-\frac{n}{T}\tau}d\tau \tag{10}$$

$$= k(\frac{T}{n})^{n+1} \left[ \gamma(n+1, \frac{n}{T}t) - \gamma(n+1, \frac{n}{T}\max(0, t-1)) \right]$$

where $\gamma$ is the incomplete Gamma function.

## B.2 SPARSIFICATION

In matrix form, sparsification with indices in $\mathcal{T} = \{t_1, ..., t_k\}$ then corresponds to a matrix multiplication with the diagonal matrix $\mathbf{S}_{\mathcal{T}} = \text{diag}(\mathbf{1}_{\mathcal{T}})$,

$$\tilde{\mathbf{x}}[\mathbf{t}] = \mathbf{S}_{\mathcal{T}}\mathbf{x}[\mathbf{t}]$$

When using the Hebbian-like term $\mathbf{h}[t] = f'(\mathbf{x}[t]^\top \mathbf{w})\mathbf{x}[t]$ as inputs to the SSM, the gradient computation when both inputs and gradients are sparsified, respectively with $\mathcal{T}, \mathcal{T}'$ will be

$$\frac{\partial L(\mathbf{x}[t]^\top \mathbf{w})}{\partial \mathbf{w}} = \mathbf{S}_{\mathcal{T}'}\delta[t] \odot \mathbf{G}\mathbf{S}_{\mathcal{T}}\mathbf{h}[t]$$

$$= \mathbf{S}_{\mathcal{T}'}\delta[t] \odot \mathbf{S}_{\mathcal{T}'}\mathbf{G}\mathbf{S}_{\mathcal{T}}\mathbf{h}[t]. \tag{11}$$

Since the sparsifying matrices are indicator functions, this is equivalent to indexing $\delta_t$, $\mathbf{G}$, and $\mathbf{h}[\mathbf{t}]$ at the appropriate positions defined by $\mathcal{T}$ and $\mathcal{T}'$. When simulating sparsity, we obtain the original input presentation indices $\mathcal{T}$ as the salient image indices and compute the arrival time gradient indices as $\mathcal{T}' = \mathcal{T} + (m-1)T$. The gradient over the batch is then computed by summing over the time—or equivalently, batch—dimension.

## C EXPERIMENTAL DETAILS

**Visual experiments.** For visual experiments, we consider the batch dimension to be the time dimension, and we compute the delayed signals over the batch dimension using a matrix convolution as described in Section B.2. The experiments in Section 4.1 use a batch size of 128 samples, while the experiments in Section 4.2 use a batch size of 1280 samples, where only the samples with the top 1.25% of training losses are used. Networks were trained using the cross-entropy loss and the AdamW optimizer with $\beta_1 = 0.9$, $\beta_2 = 0.999$. The learning rate was scaled with a linear warm-up over 10% and 20% of the training steps for Section 4.1 and Section 4.2, respectively, followed by cosine annealing to 10% of the initial learning rate. For the experiments in Section 4.1, the maximal learning rate was selected from a logarithmic grid of 5 points spanning $10^{-3}$ to $10^{-7}$, and the weight decay was chosen from the set $\{0.1, 0.01, 0.001, 0.0\}$. For the CIFAR-10 experiments in Section 4.2, the maximal learning rate was selected from $\{5 \times 10^{-5}, 2.5 \times 10^{-5}, 1 \times 10^{-5}, 7.5 \times 10^{-6}, 5 \times 10^{-6}, 2.5 \times 10^{-6}, 1 \times 10^{-6}\}$, and the weight decay was fixed to 0.1. Hyper-parameters for the MNIST experiments in Section 4.2, were searched the same way as for Section 4.1. Hyper-parameters presented in Tables 1 to 4 were independently selected using a 90%/10% split of the standard training set, and the models were retrained using the standard training set and tested on the standard test set. All experiments in Section 4.1 as well as the MNIST experiments in Section 4.2 were run for 20000 training steps. The CIFAR-10 experiments in Section 4.2 were run for 1562500 steps, in large part due to lower learning rates. Data augmentation using random horizontal flips was applied only to the CIFAR-10 experiments.

Table 1: Experiment configurations for CIFAR-10 experiments at behavioural timescales.

| CET order | delay | lr | weight decay |
|---|---|---|---|
| 1 | 0.2 | 1e-3 | 1e-1 |
| 2 | 0.2 | 1e-3 | 1e-1 |
| 6 | 0.2 | 1e-3 | 1e-1 |
| 10 | 0.2 | 1e-3 | 1e-1 |
| 1 | 0.6 | 1e-3 | 1e-3 |
| 2 | 0.6 | 1e-3 | 1e-1 |
| 6 | 0.6 | 1e-3 | 1e-1 |
| 10 | 0.6 | 1e-3 | 1e-1 |
| 1 | 1.0 | 1e-4 | 0 |
| 2 | 1.0 | 1e-3 | 1e-2 |
| 6 | 1.0 | 1e-3 | 1e-1 |
| 10 | 1.0 | 1e-3 | 1e-1 |
| 1 | 2.0 | 1e-4 | 1e-3 |
| 2 | 2.0 | 1e-4 | 1e-3 |
| 6 | 2.0 | 1e-3 | 1e-3 |
| 10 | 2.0 | 1e-3 | 0 |
| 1 | 4.0 | 1e-4 | 1e-3 |
| 2 | 4.0 | 1e-4 | 0 |
| 6 | 4.0 | 1e-4 | 1e-3 |
| 10 | 4.0 | 1e-4 | 1e-2 |
| 1 | 10.0 | 1e-4 | 0 |
| 2 | 10.0 | 1e-4 | 1e-2 |
| 6 | 10.0 | 1e-4 | 1e-3 |
| 10 | 10.0 | 1e-4 | 1e-3 |

Table 2: Experiment configurations for MNIST experiments at behavioural timescales.

| CET order | delay | lr | weight decay |
|---|---|---|---|
| 1 | 0.2 | 1e-3 | 1e-1 |
| 2 | 0.2 | 1e-3 | 1e-3 |
| 6 | 0.2 | 1e-3 | 1e-2 |
| 10 | 0.2 | 1e-3 | 1e-2 |
| 1 | 0.6 | 1e-3 | 1e-3 |
| 2 | 0.6 | 1e-3 | 1e-1 |
| 6 | 0.6 | 1e-3 | 1e-1 |
| 10 | 0.6 | 1e-3 | 1e-3 |
| 1 | 1.0 | 1e-3 | 0 |
| 2 | 1.0 | 1e-3 | 0 |
| 6 | 1.0 | 1e-3 | 0 |
| 10 | 1.0 | 1e-3 | 1e-3 |
| 1 | 2.0 | 1e-3 | 1e-3 |
| 2 | 2.0 | 1e-3 | 1e-2 |
| 6 | 2.0 | 1e-3 | 0 |
| 10 | 2.0 | 1e-3 | 1e-2 |
| 1 | 4.0 | 1e-3 | 0 |
| 2 | 4.0 | 1e-3 | 0 |
| 6 | 4.0 | 1e-3 | 0 |
| 10 | 4.0 | 1e-3 | 1e-3 |
| 1 | 10.0 | 1e-4 | 0 |
| 2 | 10.0 | 1e-4 | 1e-2 |
| 6 | 10.0 | 1e-3 | 0 |
| 10 | 10.0 | 1e-3 | 1e-3 |

Table 3: Experiment configurations for CIFAR-10 experiments at retrograde timescales.

| CET order | lr | weight decay |
|---|---|---|
| 1 | 1e-5 | 1e-1 |
| 2 | 1e-5 | 1e-1 |
| 6 | 5e-5 | 1e-1 |
| 10 | 5e-5 | 1e-1 |

Table 4: Configurations for MNIST experiments at retrograde timescales.

| CET order | lr | weight decay |
|---|---|---|
| 1 | 1e-4 | 1e-3 |
| 2 | 1e-4 | 1e-1 |
| 6 | 1e-4 | 1e-3 |
| 10 | 1e-4 | 0.0 |

**RL.** In Actor-Critic, we train the Critic using the standard $\lambda$-return, while the Actor is trained using RL eligibility traces (see Algorithm 1). The term $\nabla_\theta \log \pi_\theta(a_t \mid s_t)$ in Algorithm 1 refers either to the true gradient obtained via backpropagation or to its ET/CET approximations, computed as the product of the top-level gradient signal and the ET/CET output. The CET update is computed using Eq. 8.

The learning rate was selected from the set $2.5\mathrm{e}{-4},\ 5\mathrm{e}{-4},\ 9\mathrm{e}{-4},\ 1\mathrm{e}{-4}$ based on performance for all experiments. For all classic ET runs, the ET discount factor, $\beta$, was chosen from $0.5,\ 0.7,\ 0.9,\ 0.99$. Additionally, we used two normalization schemes for CET outputs: *area* and *peak* normalizations. In *area* normalization, the CET output is scaled so that the response to a unit input integrates to one across all future states. In *peak* normalization, the CET output is scaled such that the maximum response to a unit input is one. For Fig.8, this normalization hyperparameter was also searched.

For the Critic, we used either the same architecture as the Actor, a three-layer MLP with hidden dimension 256, or a convolutional neural network (CNN) for MinAtar/SpaceInvaders-v0, consisting of three convolutional layers (kernel size 3, zero-padding 1) followed by two fully connected layers. The ReLU activation function was used in all experiments. For CartPole and LunarLander, we also controlled the simulated time elapsed between the environment receiving an action from the agent and producing the corresponding next state and reward. This time was set to 200 ms for behavioral timescale experiments (Section 4.1) and 300 ms for retrograde signaling (Section 4.2), based on the time modeling assumptions described in the referenced sections.

---

**Algorithm 1** Actor learning via RL eligibility traces.

---

1: Initialize actor parameters $\theta$, RL eligibility trace vector $e = 0$, gradient accumulator $\nabla_\theta L = 0$, learning rate $\alpha$, and trace decay $\lambda$.
2: Sample initial state $s_0$ from the environment
3: **for** $t \in 0, \ldots, L$ **do**
4:      Select action $a_t \sim \pi_\theta(\cdot \mid s_t)$
5:      Take action $a_t$, observe $r_t, s_{t+1}$
6:      Update RL eligibility trace vector: $e = \lambda\gamma e + \nabla_\theta \log \pi_\theta(a_t \mid s_t)$
7:      Compute TD error: $\eta_t = r_t + \gamma V(s_{t+1}) - V(s_t)$
8:      Accumulate gradient: $\nabla_\theta L = \nabla_\theta L + \eta_t e$
9:      **if** $t \mod n = 0$ **then**
10:          Update actor: $\theta = \theta + \alpha \nabla_\theta L$
11:          Reset gradient: $\nabla_\theta L = 0$
12:      **end if**
13: **end for**

---

Table 5: Hyperparameters used in RL experiments.

| Parameter | Value |
|---|---|
| **Common** | |
| Optimizer | Adam |
| Adam beta | (0.9, 0.999) |
| Adam epsilon | 1e-5 |
| Weight decay | 0 |
| Policy entropy regularization coefficient | 0.01 |
| Maximum gradient norm for clipping | 0.5 |
| Learning rate | Tuned |
| Discount rate $\gamma$ | 0.99 |
| | |
| **CartPole-v1** | |
| Total number of samples | 5_000_000 |
| Number of environments | 4 |
| Number of steps to accumulate a policy gradient | 128 |
| Lambda for general advantage estimation | 0.95 |
| Anneal lr | True |
| CET normalization | Peak |
| | |
| **LunaLander-v2** | |
| Total number of samples | 5_000_000 |
| Number of environments | 4 |
| Number of steps to accumulate a policy gradient | 128 |
| Lambda for general advantage estimation | 0.95 |
| Anneal lr | False |
| CET normalization | Area or Tuned |
| | |
| **MinAtar/SpaceInvaders-v0** | |
| Total number of samples | 10_000_000 |
| Number of environments | 32 |
| Number of steps to accumulate a policy gradient | 32 |
| Lambda for general advantage estimation | Random Uniform(0.1, 0.99) |
| Anneal lr | False |
| CET normalization | Peak |

To better preserve gradient alignment with ET/CET in the first layer, we ensured positive inputs by doubling the input dimensionality and representing each original dimension with separate positive and negative components.

Table 6: Optimal learning rate and ET discounting factor configurations, $\beta$, for experiments at behavioral timescales.

**CartPole-v1**

| CET order | delay | lr | $\beta$ |
|---|---|---|---|
| backprop | - | 0.00090 | - |
| 1 | 2 | 0.00050 | 0.5 |
| 2 | 2 | 0.00025 | - |
| 5 | 2 | 0.00025 | - |
| 8 | 2 | 0.00025 | - |
| 10 | 2 | 0.00025 | - |
| 1 | 4 | 0.00050 | 0.9 |
| 2 | 4 | 0.00025 | - |
| 8 | 4 | 0.00025 | - |
| 5 | 4 | 0.00025 | - |
| 10 | 4 | 0.00050 | - |
| 1 | 8 | 0.00050 | 0.99 |
| 2 | 8 | 0.00025 | - |
| 5 | 8 | 0.00025 | - |
| 8 | 8 | 0.00025 | - |
| 10 | 8 | 0.00050 | - |
| 1 | 32 | 0.00010 | 0.99 |
| 2 | 32 | 0.00025 | - |
| 5 | 32 | 0.00025 | - |
| 8 | 32 | 0.00025 | - |
| 10 | 32 | 0.00010 | - |
| 1 | 64 | 0.00010 | 0.99 |
| 2 | 64 | 0.00025 | - |
| 5 | 64 | 0.00090 | - |
| 8 | 64 | 0.00090 | - |
| 10 | 64 | 0.00050 | - |
| 1 | 128 | 0.00050 | 0.5 |
| 2 | 128 | 0.00010 | - |
| 5 | 128 | 0.00025 | - |
| 8 | 128 | 0.00025 | - |
| 10 | 128 | 0.00025 | - |

**LunarLander-v2**

| CET order | delay | lr | $\beta$ |
|---|---|---|---|
| backprop | - | 0.00050 | - |
| 1 | 2 | 0.00090 | 0.99 |
| 2 | 2 | 0.00050 | - |
| 5 | 2 | 0.00050 | - |
| 8 | 2 | 0.00050 | - |
| 10 | 2 | 0.00050 | - |
| 1 | 4 | 0.00090 | 0.99 |
| 2 | 4 | 0.00050 | - |
| 5 | 4 | 0.00050 | - |
| 8 | 4 | 0.00050 | - |
| 10 | 4 | 0.00090 | - |
| 1 | 8 | 0.00090 | 0.99 |
| 2 | 8 | 0.00050 | - |
| 5 | 8 | 0.00050 | - |
| 8 | 8 | 0.00090 | - |
| 10 | 8 | 0.00050 | - |
| 1 | 32 | 0.00090 | 0.99 |
| 2 | 32 | 0.00090 | - |
| 5 | 32 | 0.00090 | - |
| 8 | 32 | 0.00050 | - |
| 10 | 32 | 0.00025 | - |
| 1 | 64 | 0.00090 | 0.99 |
| 2 | 64 | 0.00090 | - |
| 5 | 64 | 0.00090 | - |
| 8 | 64 | 0.00090 | - |
| 10 | 64 | 0.00090 | - |
| 1 | 128 | 0.00050 | 0.7 |
| 2 | 128 | 0.00090 | - |
| 5 | 128 | 0.00090 | - |
| 8 | 128 | 0.00090 | - |
| 10 | 128 | 0.00090 | - |

**MinAtar/SpaceInvaders-v0**

| CET order | delay | lr | $\beta$ |
|---|---|---|---|
| backprop | 1 | 0.00090 | - |
| 1 | 1.2 | 0.00025 | 0.9 |
| 2 | 1.2 | 0.00090 | - |
| 5 | 1.2 | 0.00050 | - |
| 8 | 1.2 | 0.00050 | - |
| 10 | 1.2 | 0.00090 | - |
| 1 | 2 | 0.00090 | 0.99 |
| 2 | 2 | 0.00050 | - |
| 5 | 2 | 0.00050 | - |
| 8 | 2 | 0.00050 | - |
| 10 | 2 | 0.00050 | - |
| 1 | 4 | 0.00090 | 0.99 |
| 2 | 4 | 0.00050 | - |
| 5 | 4 | 0.00090 | - |
| 8 | 4 | 0.00050 | - |
| 10 | 4 | 0.00050 | - |
| 1 | 8 | 0.00050 | 0.99 |
| 2 | 8 | 0.00050 | - |
| 5 | 8 | 0.00090 | - |
| 8 | 8 | 0.00025 | - |
| 10 | 8 | 0.00050 | - |

Table 7: Optimal learning rate, CET normalization, and ET discounting factor configurations, $\beta$, for experiments at retrograde timescales.

**CartPole-v1**

| CET order | lr | normalization | $\beta$ |
|---|---|---|---|
| 1 | 0.00010 | - | 0.9 |
| 2 | 0.00010 | peak | - |
| 5 | 0.00010 | peak | - |
| 10 | 0.00025 | peak | - |

**LunarLander-v2**

| CET order | lr | normalization | $\beta$ |
|---|---|---|---|
| 1 | 0.00025 | - | 0.5 |
| 2 | 0.00050 | area | - |
| 5 | 0.00025 | area | - |
| 10 | 0.00050 | peak | - |

The remaining hyperparameters used in the experiments are summarized in Table 5, and tuned hyperparameters are reported in Tables 6 and 7. Note that for MinAtar/SpaceInvaders, we use a randomly sampled $\lambda$ value, as we found this improves performance in this environment. A separate $\lambda$ is sampled independently for every learned parameter, which is feasible due to our RL eligibility traces implementation of Actor learning.

**Compute.** All experiments were done on RTX 8000 and A100 GPUs. Each MNIST run takes between 3 and 10 minutes on an RTX 8000 GPU, while each CIFAR-10 run takes approximately 30 minutes for Section 4.1 and up to 24 hours for Section 4.2 on a RTX 8000. Each RL run takes approximately 1-2 hours to complete.

## D SUPPLEMENTAL RESULTS ON GRADIENT ALIGNMENT

To complement the analyses in the main text, we provide additional results on gradient alignment across all layers for different tasks. First, we show in Fig. 9 that the separation observed in Fig. 5 generally holds across different layers. A similar trend is observed for RL tasks in Fig. 11, although the separation is much noisier and sometimes does not hold for the first layer. We hypothesize that the training dynamics of ET and CET can differ significantly, guiding parameters to distinct regions

in the loss landscape. In these regions, gradient alignment might occasionally be higher for ET, yet overall performance remains lower.

We can also see in Fig. 10 that the separation between the different CETs is still evident when plotting the similarity against training steps for MNIST. However, Fig. 12 shows that this relationship is muddied for CIFAR-10, which justifies plotting against accuracy.

Additionally, we note that no experiments were conducted with MinAtar/SpaceInvaders at the retrograde timescale. As shown in the rightmost heatmap of Fig. 4, CET does not scale well to longer timescales on SpaceInvaders, exhibiting only modest performance with an 8-second delay and consequently failing at a 120-second delay at the retrograde timescale (not shown).

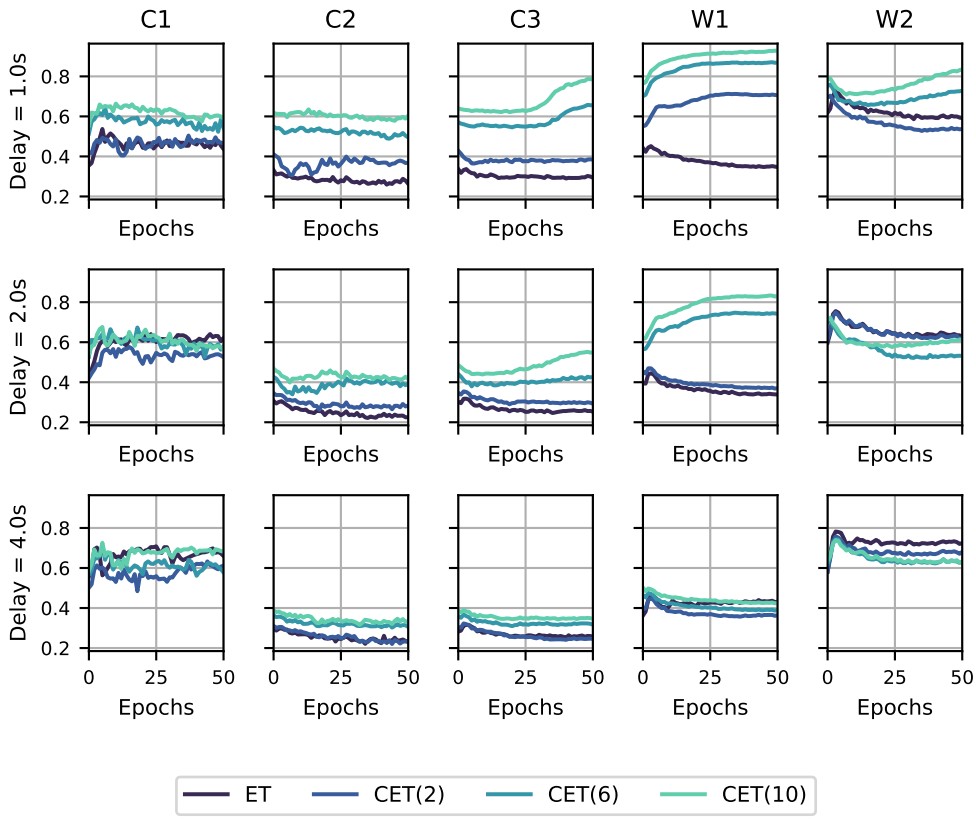

Figure 9: Cosine similarity for different layers of a CNN between between true gradients and ETs or CETs approximated gradients for all considered environments during training on CIFAR-10. C1-3: convolutional layers 1-3; W1-2: MLP layers 1-2.

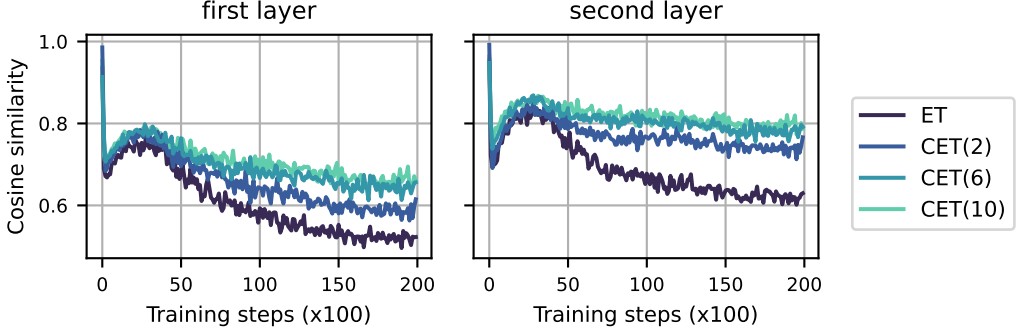

Figure 10: Cosine similarity between the true gradient and the ET/CET approximation in a retrograde setting. Each plot shows an individual layer of an MLP during training on MNIST

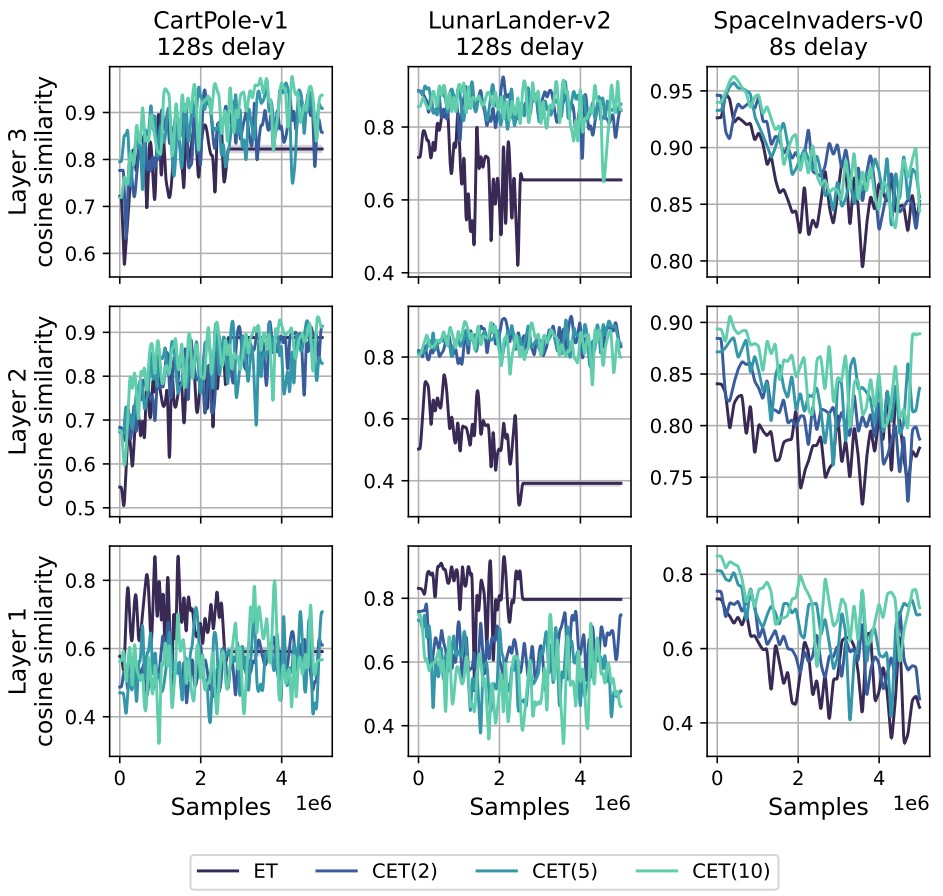

Figure 11: Average cosine similarity between the true gradients and those approximated by ETs or CETs during training, computed for each layer of a 3-layer MLP across all considered environments. The delay is set to the maximum behavioral-timescale value reported in the main text.

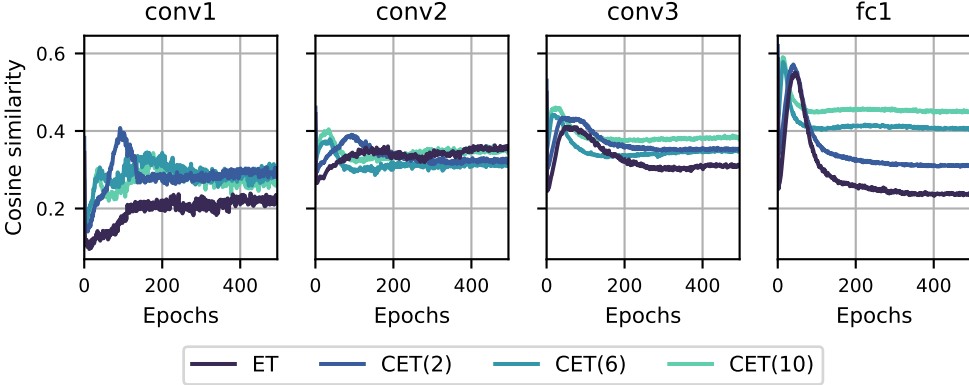

Figure 12: Cosine similarity between the true gradient and the ET/CET approximation in a retrograde setting. Each plot shows an individual layer of a CNN during training on CIFAR-10

# E    SUPPLEMENTAL RESULTS WITH RESERVOIR (RECURRENT) NEURAL NETWORK

To demonstrate the compatibility of CETs with learning in recurrent settings, we consider a partially observable variant of LunarLander-v2 in which one velocity component is masked (POMDP). In this setting, memory improves performance, motivating recurrent augmentation.

We employ a reservoir network, a recurrent architecture with a *fixed*, non-trainable recurrent matrix (commonly used in motor learning; see, e.g., Sussillo & Abbott (2009); Hoerzer et al. (2014)). Concretely, we augment the masked POMDP observation with a 256-dimensional reservoir state whose dynamics are driven by the raw LunarLander-v2 observation. Apart from this augmentation, the downstream MLP head is identical to the feedforward architecture used in the fully observable MDP. Both the feedforward and reservoir-augmented variants are trained with CET(5) under a fixed delay of 2 seconds.

CETs effectively solved the POMDP task (average return $> 200$) with the reservoir augmentation, whereas a non-recurrent MLP baseline failed entirely (average return $\approx 0$), see Table 8. For completeness, we also evaluated Exponential Traces (ET) on the same reservoir architecture with decay factors $0.5$ and $0.9$; both configurations failed to learn (average return $\approx 0$).

Table 8: Partially observable LunarLander-v2 with masked velocity and 2s. delay. Reservoir has 256 units with fixed recurrent weights.

| Architecture / Rule | Avg. Return |
|---|---|
| MLP (no recurrence) + CET(5) | $\approx 0$ |
| Reservoir (256) + ET | $\approx 0$ |
| **Reservoir (256) + CET(5)** | $> 200$ |

In a POMDP with delayed credit assignment, CETs leverage recurrent state to overcome temporal mismatch, while ET fails under identical conditions.

# F    TINYIMAGENET

As a step towards scalability, we report results on TINY IMAGENET (Table 9) under the same training setup as Section 4.1, using a *ResNet-20* with strided convolutions replaced with average pooling. We compare Exponential Traces (ET), Cascading Eligibility Traces with 10 states, and standard backpropagation. We evaluate performance with a fixed delay of 1 second after 50K training steps.

Table 9: TINY IMAGENET performance with a $1\,\mathrm{s}$ delay after 50K training steps.

| Method | Top-1 Acc. | Top-5 Acc. |
|--------|------------|------------|
| ET | 0.0973 | 0.2658 |
| CET (10) | 0.3431 | 0.6136 |
| Backprop | 0.4060 | 0.6791 |

These results reinforce our hypothesis that more complex visual tasks are more sensitive to temporal mismatch, and that CETs provide substantial gains when the system is well matched to the delay. To check whether ET could be improved by mitigating gradient alignment issues via a smaller learning rate, we also tested ET with a reduced learning rate; for equal training time (50K steps), but the performance decreased relative to the table above.

## G APPROXIMATING ACTIVATION FUNCTION DERIVATIVES ON RETROGRADE TIMESCALE

In Subsection 4.2, for retrograde experiments to test the CET model ability with fewer compounding factors, we assumed a simplified setup: perfect derivatives of activation functions are given to us by an oracle, though they are delayed in the same way as activations are. In this supplementary section, we remove this assumption.

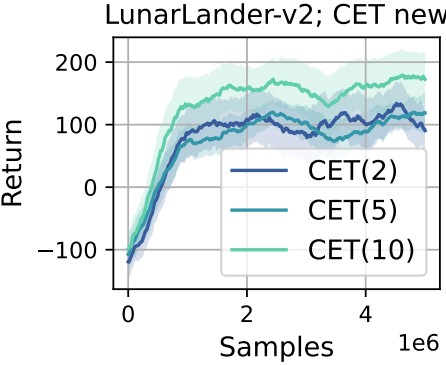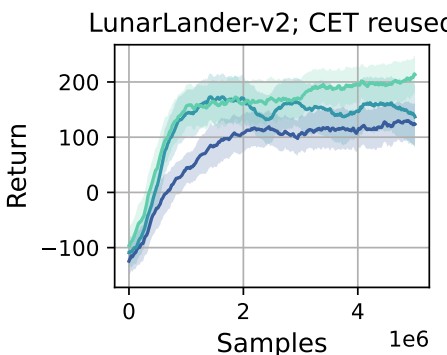

Figure 13: Preliminary results on approximating activation function derivatives on a retrograde a 2-minute delay setup. **Left:** We use an independent additional CET model to approximate the derivative of the activation function. **Right:** We use the same CET model that we use to approximate activations to also approximate the derivatives. In both cases, we use a CET output threshold (*CET output > threshold*) to obtain the final approximation, motivated by the usage of a ReLU activation function.

To account for delayed activation function derivatives, we perform preliminary experiments with two setups: (1) we use a separate CET model to approximate the derivatives during the backward pass; (2) we reuse the same CET model that we use to approximate activations at each layer. Given that we use a ReLU activation function, to obtain the final derivative approximation we apply a threshold on the CET output: *CET output > threshold*. Results are shown in Fig. 13.

We used learning rate of 5e-4 and the threshold value was set to 1 in all experiments in this section. This choice of the threshold is motivated by the fact that in all LunarLander experiments the CET output is scaled such that the maximum response to a unit input is one. We view this as a reasonable heuristic, although more work is needed to derive an optimal threshold value, and the results should be considered preliminary.

## H  VARYING THE ALPHA PARAMETER FOR A FIXED DELAY

In Section 3, we assume that the parameter $\alpha$ is chosen as $\alpha = \frac{n-1}{T}$, which peaks at $t = T$. In this section, we assume the delay is $T$, but we vary $\alpha = \frac{n-1}{T'}$ by varying $T'$, making it peak at the wrong $T'$. We experiment with a 32-second delay, $T$, on the behavioral timescale in LunarLander, varying the CET peak, $T'$, from 8 seconds to 56 seconds. Fig. 14 shows the results.

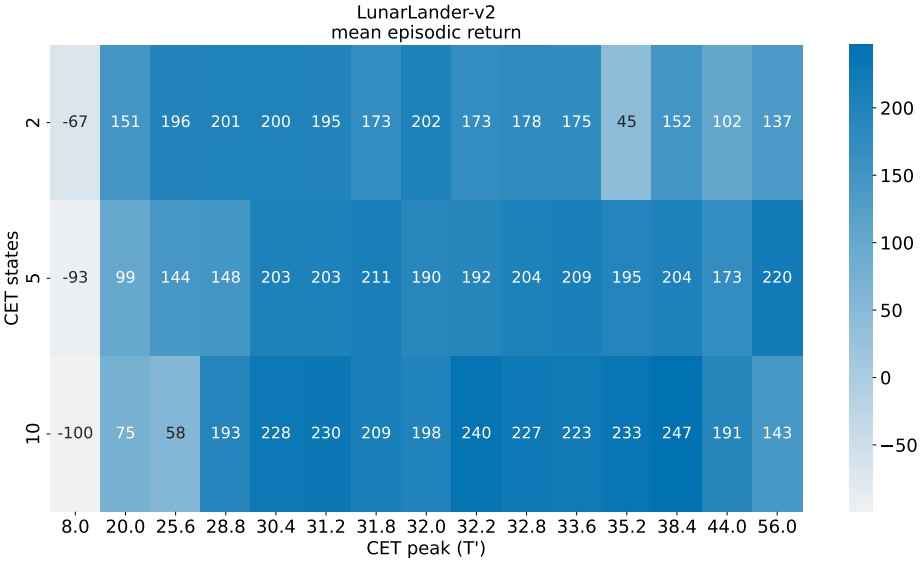

Figure 14: Mean episodic return for the LunarLander environment with a delay of 32 seconds across different numbers of CET states and $\alpha$ parameters that peak at shown timesteps. All results are averaged over 3 seeds. Note when $T'$ is equal to 32 seconds it recovers a kernel shape assumed in the rest of the paper.

While the results are noisy, we can see that $T'$ can vary by a few seconds without hurting performance; it is not symmetric as it supports overestimating the delay more than underestimating it, which is consistent with the shape of the kernel. Moreover, performance degrades more rapidly for narrower kernels (e.g., CET(10)), as expected.

Similar to the experiments in the main text, the CET output is scaled such that the maximum response to a unit input is one, which is achieved at time $T'$. We used a fixed learning rate of 9e-4. That and different seeding explains the difference between these results and the results in the main paper.

## I  NON I.I.D. VISUAL DATA

To study the effect of non i.i.d. visual streams on performance, we conduct additional experiments on synthetic videos derived from CIFAR-10 in a setup matching experiments in Section 4.1. We generate temporally coherent sequences by interpolating intermediate frames and the label probabilities using a Hahn kernel. Across interpolation factors, the proposed approach maintains stable performance on these non-i.i.d. inputs. Intuitively, increasing temporal correlation shifts the input spectrum toward lower frequencies, where CETs are particularly effective because they act as low-pass temporal filters whose fidelity improves with $n$.

## J  SUPPLEMENTAL RESULTS WITH LEAKY INTEGRATE-AND-FIRE NEURONS

As an additional step towards biologically plausible learning, we evaluate a feedforward network of leaky integrate-and-fire (LIF) neurons on the standard LunarLander-v2, see Table 10. We compare ET and CET with 5 states across multiple feedback delays. LIF neurons are implemented with

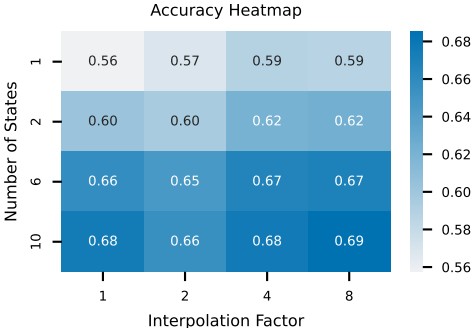

Figure 15: Accuracy for CIFAR-10 dataset across varying numbers of CET states and interpolation factors with delay of 2s. Interpolation factor of 1 corresponds to i.i.d. setting, while factors 2,4, and 8 introduce 1,3, and 7 interpolated frames between real images respectively during training. Performance is measured on non-interpolated frames only.

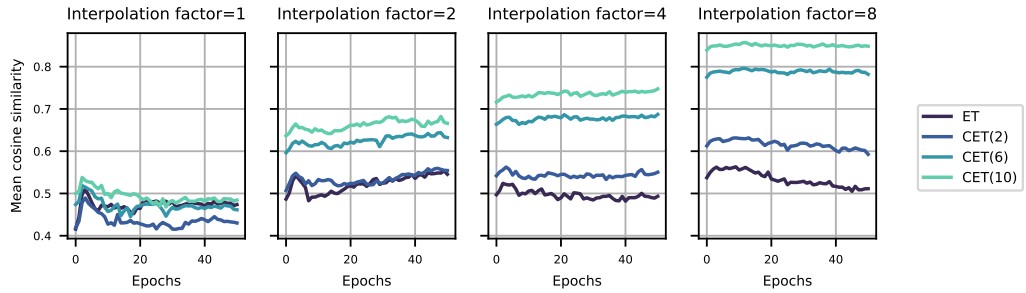

Figure 16: Average cosine similarity between the true gradient and the gradients produced by CETs on CIFAR-10, evaluated across varying numbers of CET states and interpolation factors with a fixed delay of 2 s.

`snnTorch`, using membrane decay $\beta = 0.9$ and a fast-sigmoid surrogate gradient for the spike nonlinearity.

Table 10: LunarLander-v2 average return with LIF neurons across delays (in seconds).

| Method / Delay (s) | 2 | 8 | 16 | 32 | 64 |
|---|---|---|---|---|---|
| ET | 274 | 226 | $-133$ | $-136$ | $-89$ |
| CET(5) | 270 | 246 | 127 | 122 | 64 |

The results mirror the trends in the main paper: increasing delay degrades performance for both ET and CET, but CET becomes significantly better than ET at longer delays (e.g., $T \geq 16s$).

All architectural components other than the spiking layer remain identical to the feedforward MLP used in the main text.

## K    STANDARD DEVIATION FOR REINFORCEMENT LEARNING RESULTS

We report results with standard deviations for RL tasks reported in the main text at the behavioral timescale in Fig. 17 to show reliability.

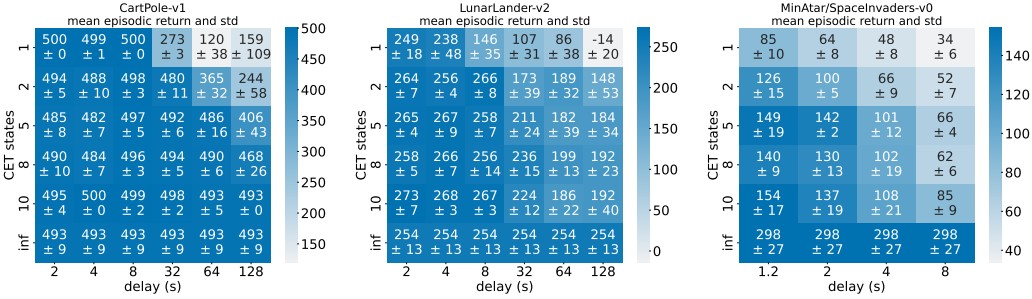

Figure 17: Same as Fig. 4, but with standard deviation over three seeds. "inf" refers to backpropagation baseline.

## L  SUPPLEMENTAL RESULTS ON VARIABLE DELAY

We report result on variable delay in three settings long delays, moderate delays, and to the RL.

Across settings, CETs consistently outperform ETs for unimodal, peaky variable–reward/feedback distributions. As expected, when the delay variance approaches the full delay range (i.e., the truncated Gaussian approaches uniform), the performance gap between CET and ET diminishes.

### L.1  VARIABLE DELAYS ON BEHAVIORAL TIME SCALE

For visual experiments, we study robustness to variable feedback delays by drawing per-trial delays from a Gaussian centered at the nominal mean ($T = 1\,\mathrm{s}$ or $T = 2\,\mathrm{s}$), truncated to $[0, 2T]$. We sweep standard deviations $T_{\mathrm{var}} \in \{0, \frac{1}{4}T, \frac{1}{2}T, T\}$, mirroring the CIFAR-10 setup in Section 4.1.

CETs consistently outperform ETs under variable delays; increasing CET degree improves accuracy even when $T_{\mathrm{var}} = \frac{1}{2}T$. Importantly, CET performance does not collapse below ET under high variability.

Table 11: Accuracy vs. standard deviation for $T = 1\,\mathrm{s}$ with delays in $[0, 2T]$.

| Method | $T_{\mathrm{var}}=0.00$ | $T_{\mathrm{var}}=0.25$ | $T_{\mathrm{var}}=0.50$ | $T_{\mathrm{var}}=1.00$ |
|---|---|---|---|---|
| ET | 0.68 | 0.68 | 0.68 | 0.68 |
| CET(2) | 0.71 | 0.70 | 0.70 | 0.68 |
| CET(6) | 0.73 | 0.73 | 0.72 | 0.70 |
| CET(10) | 0.74 | 0.73 | 0.72 | 0.69 |

Table 12: Accuracy vs. standard deviation for $T = 2\,\mathrm{s}$ with delays in $[0, 2T]$.

| # States | $T_{\mathrm{var}}=0.00$ | $T_{\mathrm{var}}=0.50$ | $T_{\mathrm{var}}=1.00$ | $T_{\mathrm{var}}=2.00$ |
|---|---|---|---|---|
| ET | 0.62 | 0.63 | 0.64 | 0.64 |
| CET(2) | 0.66 | 0.66 | 0.65 | 0.63 |
| CET(6) | 0.71 | 0.70 | 0.69 | 0.65 |
| CET(10) | 0.72 | 0.72 | 0.70 | 0.65 |

### L.2  VARIABLE DELAYS ON MINUTES-SCALE

We also evaluate CIFAR-10 with delays sampled from a truncated Gaussian on $[0, 2T]$ at minutes-scale and show that CETs remain viable under longer delays with sparse activity. In these experiments, we consider a variable delay applied uniformly to the entire network and the sparsification method in Section B.2 used in Section 4.2 with random 1% sparsity.

Table 13: CIFAR-10 accuracy with mean delay $T = 60\,\mathrm{s}$ and range $[0, 2T]$.

| Method | $T_{\mathrm{var}}$=0.00 | $T_{\mathrm{var}}$=15.00 | $T_{\mathrm{var}}$=30.00 | $T_{\mathrm{var}}$=60.00 |
|---|---|---|---|---|
| ET | 0.70 | 0.70 | 0.71 | 0.69 |
| CET(2) | 0.73 | 0.72 | 0.72 | 0.70 |
| CET(6) | 0.74 | 0.74 | 0.72 | 0.69 |
| CET(10) | 0.74 | 0.74 | 0.72 | 0.68 |

Table 14: CIFAR-10 accuracy with mean delay $T = 120\,\mathrm{s}$ and range $[0, 2T]$.

| Method | $T_{\mathrm{var}}$=0.00 | $T_{\mathrm{var}}$=30.00 | $T_{\mathrm{var}}$=60.00 | $T_{\mathrm{var}}$=120.00 |
|---|---|---|---|---|
| ET | 0.67 | 0.68 | 0.68 | 0.65 |
| CET(2) | 0.70 | 0.69 | 0.68 | 0.66 |
| CET(6) | 0.73 | 0.72 | 0.70 | 0.66 |
| CET(10) | 0.73 | 0.73 | 0.70 | 0.65 |

### L.3 Variable Delay on RL tasks

We report variable feedback delay for LunarLander-v2 under a mean delay of $T = 32s$ and different trancated variances in Table 15.

Table 15: Mean episodic return on LunarLander-v2 under a mean delay of $T = 32s$, across different delay variances.

| Method / Var | var = 2.0 | var = 8.0 | var = 16 | var = 32 |
|---|---|---|---|---|
| ET | $-42$ | $-66$ | $-146$ | $-152$ |
| CET(5) | **198** | **55** | **81** | $-109$ |

### L.4 Variable Delays: Unknown Fixed Delay (Learned)

We next consider fixed but *unknown* delays that must be learned. The CET kernel is initialized to peak at $200\,\mathrm{ms}$ (ET initialized with $200\,\mathrm{ms}$ mean as an exponential distribution). We tune a single delay parameter $\alpha$ via a simple weight-perturbation (finite-difference) update:

$$\Delta\alpha \;=\; -\,\eta\,\frac{L_{+} - L_{-}}{2\delta},$$

where $\delta \sim \mathcal{N}(0, 1)$, and $L_{+}$ (resp. $L_{-}$) is the loss obtained when computing the network update assuming time constant $\alpha+\delta$ (resp. $\alpha-\delta$). This approximates gradient descent on $L(\cdot)$ in expectation via a first-order Taylor expansion.

We can see that eligibility trace performance degrades if the decay parameter is poorly initialized, more so for higher-order CETs; thus time-constant tuning is a shared challenge. Nevertheless, when $\alpha$ is learned online, CETs outperform ETs.

Table 16: Accuracy vs. true (unknown) delay $T$ when initialized at $200\,\mathrm{ms}$.

| # States | $T$=1 s | $T$=2 s |
|---|---|---|
| ET | 0.67 | 0.55 |
| CET(2) | 0.72 | 0.61 |
| CET(6) | 0.74 | 0.65 |
| CET(10) | 0.74 | 0.67 |

