# OpenReview forum: "Learning From the Past with Cascading Eligibility Traces"
_ICLR.cc/2026/Conference — ICLR 2026 Poster_

### Official Review · Reviewer_bZyU · 2025-10-29

**Soundness:** 3
**Presentation:** 4
**Contribution:** 3
**Rating:** 6
**Confidence:** 3

**Summary:**

This paper introduces the Cascade Eligibility Trace (CET), a novel mechanism for the temporal credit assignment problem in deep neural networks, motivated by biological plausibility. The core idea replaces the standard exponential decay of classic eligibility traces (ETs) with a cascade of $n$ coupled differential equations. The CET's primary function is to generate a memory kernel that is centered around the time of the error signal, rather than that of the sensory input. This allows learning systems to accurately assign credit for rewards or feedback signals that arrive significantly after the causal neural activity. This is achieved by effectively substituting the standard exponential kernel of ETs with a Gamma-like kernel which peaks at a given delay. The authors conduct extensive benchmarking against the classical ET across various tasks (both supervised and RL) and datasets (e.g. MNIST, LunarLander), systematically demonstrating the computational advantages of the CET paradigm. The work is well-written, with a clear and compelling narrative.

**Strengths:**

- ⁠The new formulation is clear and straightforward, addressing with clarity the weaknesses of previous models. Recasting eligibility as a Gamma-shaped temporal window with tunable peak and width is a principled generalization beyond two-exponential composites.
- Theoretical derivations for both continuous-time and exact discrete-time update. The parameters $(n, \alpha)$ have a nice interpretation as "delay" and "precision".
- ⁠The benchmarking is extensive, and most of the results are quite solid, demonstrating  a clear advantage w.r.t. classic ET, with systematic sweeps over delays and CET orders. Author provide gradient-alignment analysis to connect performance with optimization geometry.
- ⁠The application with accumulating delays is interesting from both a computational and a neuroscientific perspective.

**Weaknesses:**

- While the work strongly underlines the method’s biological plausibility, benchmarks focus on accuracy on classification and reinforcement learning tasks, but misses benchmarking against real biological data (e.g. neural predictivity metrics (e.g. Yamins et al., 2014; Yamins & DiCarlo, 2016). I’d like to see how a model trained with CET  on a image classification task (e.g. CIFAR10) compares to the same model trained with ET or backpropagation in their neural predictivity.
- The paper argues that two-ET composites produce broad windows and thus underperform for long delays. However, no optimized two-exponential (or multi-exponential) baseline is reported in the main text under the same tuning budget as CETs. Author could for example include a two-ET case with grid-searched $(\gamma_P,\gamma_D)$.
- CETs increase state size linearly in $n$, which results in per-step computational overhead. The paper should quantify the relevance of such overhead for example by comparing it with ET (e.g. FLOPs/memory). This might become relevant for example in the long-delay experiments that rely on sparsification.
- In stacked-delay experiments, the $\delta$-computation assumes access to $f’(⋅)$ at the appropriate delayed postsynaptic state (a soma-side memory).
-  It seems to me that one of the key advantages of CET is built on the assumption that the variable delays are clustered. As shown in Section I in the Appendix, the benefit of CET w.r.t. ET performances decreased as the variance increased and the delay distribution became closer to uniform. The solution suggested (Perturbation-Based Learning) violates the principle of locality, which is foundational to biological functioning. This method requires a synapse to access, store, and compare the global loss from multiple non-local, trial-level executions.

**Questions:**

1. How sensitive are results to the trade-off $T=(n-1)/\alpha$? For fixed $T$, what is the optimal $n$ given compute/memory budgets? Please provide heatmaps of performance and cosine similarity over $(n,\alpha)$ at one or two T values.
2. In stacked delays, how would you maintain the postsynaptic derivative at the soma? Could an analogous CET run there, and do you expect additional degradation?
3. What happens when the critic is also trained with delayed signals (e.g., delayed TD errors)? Does CET still yield stable learning?
4. How does CET compare with *tuned* two-exponential kernels?

---

> ### Author Response · Authors · 2025-11-26
>
> Thank you for the review! Here, we summarize the new results, and answer your questions.
>
> > The paper argues that two-ET composites produce broad windows and thus underperform for long delays. However, no optimized two-exponential (or multi-exponential) baseline is reported in the main text under the same tuning budget as CETs. Author could for example include a two-ET case with grid-searched.
>
> > How does CET compare with tuned two-exponential kernels?
>
> We appreciate the reviewer’s suggestion to strengthen the connection with prior work on multi-trace models. In response, we have added a theoretical comparison with the two trace LTP-LTD eligibility trace formulation. In Appendix A3 (Proposition 2), we study this kernel as a probability density function and show that **CET(2) is optimal: for a fixed peak, the optimal choice of parameters (a,b) that minimizes the second moment around T converges pointwise and in L1 to the CET(2) kernel**. While it would also be possible to consider the sum of more exponentials and sign-changing kernels, such an extension drastically increases the dynamical system’s complexity in terms of parameters. In contrast, the proposed approach offers a way to scale up the number of states and improve the temporal resolution while keeping a single parameter $\alpha$ that fully specifies the kernel for a fixed $n$.
>
> > In stacked-delay experiments, the -computation assumes access to  at the appropriate delayed postsynaptic state (a soma-side memory)
>
> > In stacked delays, how would you maintain the postsynaptic derivative at the soma? Could an analogous CET run there, and do you expect additional degradation?
>
> Thank you for your question! One way to approximate this derivative on a slow scale is discussed in (Fan & Mysore, 2024, specifically for retrograde models) and is based on E/I balance. Another option is to use CETs to store the postsynaptic derivative history so that it can be multiplied with upstream error signals propagating backwards. To explore this, we ran two experiments: using another CET (of the same order) to approximate the derivative, and reusing the original CET to estimate it (as this CET stores the pre-post product, it should be small when the derivative is zero), please see Appendix G. In these preliminary experiments, we obtained slightly worse performance compared to the oracle case but CET(10) still solves the environment, and the dependence on CET order behaves as expected: performance decreases for lower orders. See Appendix G for the full results.
>
> Regarding future work, because the derivative signal is shared across all presynaptic neurons, it can be stored in the postsynaptic soma. We assume that the soma would have a much larger capacity than individual synapses (the location where we currently model CETs). Future work could explore performance gains from larger, but not perfect, CET capacity in the soma.
>
> > How sensitive are results to the trade-off $T=(n-1)/\alpha$? … …Please provide heatmaps of performance and cosine similarity over $(n, \alpha)$ at one or two T values?
>
> Thank you for the suggestion! We interpreted your question as follows: *for a fixed delay and a fixed CET order, how sensitive is the performance to the hyperparameter alpha?*
>
> We ran experiments with a 32-second delay in the LunarLander environment, varying alpha so that the CET peak ranges from 8 to 56 seconds (instead of 32 seconds as in our main experiments). The results show that the CET peak can vary by a few seconds without hurting performance; the dependence on peak is not symmetric and supports overestimating the delay more than underestimating it, which is consistent with the shape of the kernel. Moreover, performance degrades more rapidly for narrower kernels (for example, CET(10)), as expected. We also found cosine similarity unusually noisy in this setting (i.e. some of low-performing runs show high cosine similarity between CET and backprop). See Appendix H for the full results and the corresponding figure.

---

> > ### Author Response · Authors · 2025-11-26
> >
> > > What happens when the critic is also trained with delayed signals (e.g., delayed TD errors)? Does CET still yield stable learning?
> >
> > We expect the results to become worse, although we have not tested this configuration, as it would likely produce noisier outcomes that are more difficult to interpret. We believe the proposed method should first be further developed before being applied to the joint actor-critic setting.
> >
> > > CETs increase state size linearly in , which results in per-step computational overhead. The paper should quantify the relevance of such overhead for example by comparing it with ET (e.g. FLOPs/memory) (Shared answer as for reviewer kyjJ)
> >
> > Implementing CET(n), with ET corresponding to CET(1), online requires maintaining and updating n state variables at every time step for each synapse. Each update consumes a constant number of operations per variable (one multiplication and one addition). As a result, the compute and memory requirements scale linearly with n. We updated the paper accordingly, see lines 199-201.
> >
> > > For fixed T, what is the optimal n given compute/memory budgets?
> >
> > A regular K×K multiplication on a K-dimensional vector comes at a cost of approximately 2K^2 operations. CET(n) adds an overhead of  2nK^2, since we need to perform 2n operations per weight connection. Thus, the total number of operations per layer for CET(n) is on the order of (2 + 2n)K^2. The memory overhead is on the order of nK^2 per layer.
> >
> > Although we can compute the overhead, the optimal n for a fixed compute/memory budget depends on the task and the delay distribution. We don’t believe it has a closed-form solution. Empirically, for easy tasks with small delays, n=2 achieves the same performance as higher orders while using less compute and memory, so it is effectively Pareto-optimal. For harder tasks with long delays, higher orders may be required to enable any learning.
> >
> > We would also like to clarify that our primary goal in this work is not to propose CET as a production-ready algorithm optimized for specific hardware constraints, but rather to test a biologically motivated mechanism for delayed credit assignment in a deep-learning setting. Since we believe it can be implemented in the brain via biochemical cascades (with n representing the number of stages in the cascade; see the Discussion section, the paragraph mentioning ETs and CaMKII), our focus is on its qualitative behavior rather than on deriving an optimal n under particular compute/memory budgets.
> >
> > We also want to note that, in our RL setup and current implementation, the wall-clock time is the same for networks with considered CET orders, as the main latency comes from interaction with the environment.
> >
> > > It seems to me that one of the key advantages of CET is built on the assumption that the variable delays are clustered. As shown in Section I in the Appendix, the benefit of CET w.r.t. ET performances decreased as the variance increased and the delay distribution became closer to uniform.
> >
> > This is indeed correct. CET is built on the assumption that the delay is unimodal (as supported by experimental evidence: see citations in lines 84 and 88).  If the delay is uniform, CET will not give an advantage over ET. Roughly speaking, the distribution of the delay should be close to the kernel shape to make this kernel advantageous compared to another kernel. For this reason, CET has an advantage when the delay is clustered (unimodal). If the distribution is uniform on the interval [0, N], the ideal kernel is likewise uniform; when ET (with a high decay factor) or CET (with a low order) approximates this uniform kernel closely enough, their performance is roughly equivalent. Finally, ET has an advantage when the delay is exponentially distributed (i.e. when it matches the shape of the ET kernel). We updated the text to clarify this point, see lines 334-335.
> >
> >  > (Perturbation-Based Learning) violates the principle of locality, which is foundational to biological functioning. This method requires a synapse to access, store, and compare the global loss from multiple non-local, trial-level executions.
> >
> > Thank you for the critical remark. We believe perturbation-based learning of alpha serves as a proof of concept that alpha can be learned with simple learning rules (in particular, with very high-variance ones such as weight perturbation). One can imagine, for example, policy gradient based learning of alpha or other more biologically plausible learning methods that respect locality.

---

> > > ### Author Response · Authors · 2025-11-26
> > >
> > > >  ...misses benchmarking against real biological data (e.g. neural predictivity metrics (e.g. Yamins et al., 2014; Yamins & DiCarlo, 2016). I’d like to see how a model trained with CET on a image classification task (e.g. CIFAR10) compares to the same model trained with ET or backpropagation in their neural predictivity.
> > >
> > > We agree that such a comparison would be interesting, although we do not expect the results to differ much (if at all) from backpropagation-trained networks. The main focus of this work was to show that CETs can provide a good enough approximation to error signals required by various learning algorithms. As such, we do not change the underlying algorithm or network architecture, which would lead to CETs producing similar representations as in backprop-trained networks.

---

### Official Review · Reviewer_7gVv · 2025-10-30

**Soundness:** 3
**Presentation:** 3
**Contribution:** 2
**Rating:** 6
**Confidence:** 3

**Summary:**

This paper introduces Cascading Eligibility Traces (CETs), a generalization of classical eligibility traces for handling delayed credit assignment in neural networks. The authors frame CETs as a state-space model inspired by biochemical cascades, which creates a temporally precise memory trace that peaks at a specific, non-zero delay. Their mechanism overcomes several of the limitations of the exponential decays in classic ETs. The paper provides extensive empirical evidence showing that CETs enable effective learning in both supervised and reinforcement learning settings with delays ranging from seconds to minutes, and connects these findings to the viability of slow retrograde axonal signaling for credit assignment in neuroscience.

**Strengths:**

The paper is well-written, and the theoretical derivations are clear and appear to be correct. The paper provides a solid and well-motivated extension of prior theory on ETs.

1. The formulation of CETs as a state-space model is mathematically sound and provides a powerful generalization of a classic learning mechanism. Showing that a standard eligibility trace is equivalent to a CET with a single state (n=1) makes this work a direct and solid extension of prior work. The model's parameters (n and alpha) offer intuitive control over the precision and timing of the memory trace.

2. They demonstrate the superiority of CETs over standard ETs across a diverse set of non-trivial supervised and RL tasks. As task complexity and feedback delay increase, the performance gap between higher-order CETs and standard ETs widens significantly. The included gradient alignment analysis provides a convincing explanation for these performance gains. I really appreciate this level of benchmarking, which is not common in basic computational neuro work.

3. Modeling learning with minute-long, stacking delays is, to my knowledge, novel and impactful. It provides a plausible computational mechanism by which slow chemical signals, such as retrograde messengers, could effectively transmit credit information.

**Weaknesses:**

1. The retrograde experiments on visual tasks rely on a salience signal to sparsify memory storage, which may be a critical component for making learning tractable over minute-long delays. If my understanding is correct, this assumes the existence of an oracle-like signal that tells the synapse which events are important and thus which memories to form, which seems to be solving part of the most difficult part of the temporal credit assignment problem (i.e. what to remember) before the delayed error signal arrives.

2. The results clearly show that more states are better. However, this benefit presumably comes at a cost. It would strengthen the paper to include a brief discussion on the computational complexity of the CET mechanism as a function of the number of states n.

**Questions:**

1. Could the authors expand on the biological plausibility of the salience signal mechanism in the main text?  A brief discussion on how a salience signal could be computed and propagated quickly enough to gate the slow memory formation process would be helpful for readers to understand the complete proposed system.

2. The theoretical derivations could possibly benefit from including the inverse Laplace transform. The kernel g(t) of the convolution in Eq 5 is the inverse Laplace transform of the function G(s)=(s+alpha)^-n. This interpretation could provide a deeper understanding of why and how decodability improves with higher n. See [Masset, Tano et al Nature 2025] for an example of using the inverse laplace transform framework to understand what kind of information is decodable from a multi-timescale system.

---

> ### Author Response · Authors · 2025-11-26
>
> Thank you for the review! Here, we summarize the new results, and answer your questions.
>
> > The results clearly show that more states are better. However, this benefit presumably comes at a cost. It would strengthen the paper to include a brief discussion on the computational complexity of the CET mechanism as a function of the number of states n.
>
> > The theoretical derivations could possibly benefit from including the inverse Laplace transform. The kernel g(t) of the convolution in Eq 5 is the inverse Laplace transform of the function G(s)=(s+alpha)^-n...
>
> We thank the reviewer for the helpful suggestion. We have added a subsection in the appendix discussing the Laplace/Fourier domain behavior of the CET(n) kernel. In particular we show that the Laplace transform of CET(n) converges pointwise to $e^{-Ts}$, the Laplace transform of the ideal delay. In particular, it implies that the kernel acts as a low-pass filter that can capture low-frequency information and that increasing n increases the filter’s threshold. This agrees with the time domain analysis where increasing $n$ concentrates the kernel around its peak, which improves decodability.
>
> However, it comes at the cost of increasing the number of states. Implementing CET(n) online requires maintaining and updating n state variables at every timestep for each synapse. Each update takes a constant number of operations per variable (one multiplication and one addition). As a result, the computation and memory requirements scale linearly in n. From a biological perspective, the SSM representation models a biochemical cascade where each state would correspond to a chemical species. Therefore, an increasing n increases the complexity of the cascade and the potential energy or material costs required to implement it biochemically.

---

> ### Author Response · Authors · 2025-11-26
>
> > The retrograde experiments on visual tasks rely on a salience signal to sparsify memory storage, which may be a critical component for making learning tractable over minute-long delays. If my understanding is correct, this assumes the existence of an oracle-like signal that tells the synapse which events are important...
>
> > Could the authors expand on the biological plausibility of the salience signal mechanism in the main text? A brief discussion on how a salience signal could be computed and propagated quickly enough to gate the slow memory formation process would be helpful for readers to understand the complete proposed system.
>
> We appreciate the reviewer’s observation regarding the role of salience-based sparsification in our experiments and agree that it is an important additional assumption. We have therefore changed the wording to make the role of sparsification more explicit (lines 361-369).
>
> Additionally, we added a paragraph in the discussion highlighting that this saliency mechanism reflects a broader class of theories in learning and predictive processing in which a subset of observations, typically those with high prediction error, drive plasticity. This idea is derives from both neuroscience and machine learning. For example, three-factor learning rules suggest that the third factor can encode surprise, effectively setting learning to zero when there is no surprise (Gerstner et al., 2018).  Large prediction error signal is also associated with greater information gain (Friston 2005; Itti & Baldi 2009; Bogacz 2020; Dabney et al. 2020), while selective consolidation of behaviourally relevant events reduces metabolic cost (van Rossum & Pache 2024).
>
> By prioritizing the top-k losses, we operationalize this principle in artificial neural networks. While we do not commit to an exact implementation for the salience signal, the role of salience gating in our model is consistent with theories in which synapses update primarily when either a global neuromodulatory factor signals high behavioural relevance (Bogacz 2020; Dabney et al. 2020), or when dendritic compartments represent  mismatches (Sacramento et al. 2017).
>
> In the context of delayed credit assignment, this assumption becomes critical because CETs must store presynaptic traces for minutes. Without prioritising high-salience events, memory buffers become saturated, and gradient estimates deteriorate. Thus, salience-based sparsification is not just computationally convenient but theoretically motivated within normative theories of learning from surprising or behaviourally important events.
>
> *References*
>
> Gerstner, Wulfram, et al. "Eligibility traces and plasticity on behavioral time scales: experimental support of neohebbian three-factor learning rules." Frontiers in neural circuits 12 (2018):
>
> Friston, Karl. "A theory of cortical responses." Philosophical transactions of the Royal Society B: Biological sciences 360.1456 (2005): 815-836.
>
> Itti, Laurent, and Pierre Baldi. "Bayesian surprise attracts human attention." Vision research 49.10 (2009): 1295-1306.
> Bogacz, Rafal. "Dopamine role in learning and action inference." Elife 9 (2020): e53262.
>
> Dabney, Will, et al. "A distributional code for value in dopamine-based reinforcement learning." Nature 577.7792 (2020): 671-675.

---

### Official Review · Reviewer_kyjJ · 2025-11-01

**Soundness:** 3
**Presentation:** 3
**Contribution:** 3
**Rating:** 6
**Confidence:** 4

**Summary:**

The paper proposes a new mechanisms for supporting temporal credit assignment in biological networks when reinforcement or error signals arrive with significant delays.The authors propose what they term as “cascading eligibility traces” (CETs), i.e., eligibility traces defined in terms of a state-space model inspired by cascading biochemical reactions. Motivated by the observation that biological systems often process error or reward signals with fixed but significant delays, the authors propose CETs to obtain temporally specific memory of past network activity, that can be matched to delayed credit signals  at behavioral time-scales ranging from seconds to minutes.

They demonstrate the proposed mechanism in a supervised experimental setting on visual classification tasks employing the MNIST/CIFAR-10 datasets (and the TinyImageNet) with delays up to seconds, where all network layers receive the same delayed error signal, on reinforcement learing tasks (namely CartPole, LunarLander, and MinAtar/SpaceInvaders), and in a retrograde signalling setting on reinforcement learning experiments, where delays stack across layers up to minutes, mimicking very slow biochemical backpropagation.

They show that increasing CET order systematically improves performance and gradient alignment when compared to standard eligibility traces (whose performance deteriorates for delayed larger than 4 seconds), but does not fully close the gap to true backpropagation for long-delay settings, where the gradient alignment stops to improve.   The authors further demonstrate results against classical eligibility traces, and provide in the supplement additional analyses that study the robustness to variable values and unknown delays, provide ablation studies on cascade order, study gradient alignment across all network layers, and application to recurrent and spiking (leaky integrate and fire) networks.
Overall the paper provides a well-motivated, biologically grounded extension of eligibility traces, however I find that the strongest biological claim depends on extra assumptions (see below).

**Strengths:**

- Clearly formulated and biologically motivated problem of wanting to model a mechanism for temporal credit assignment that is both temporally fine tuned, but also provides temporal extent to be able to account for delayed error or reinforcement signals with delays up to seconds.
- The authors provide systematic experiments both in supervised and reinforcement learning settings.
- To understand the limits in performance for long delays they study how the weight updates align with the true gradients obtained by backpropagation.

**Weaknesses:**

- There is no comparison with the eligibility trace formalisms that consider different traces for potentiation and depression (He et al., 2015; Huertas et al., 2016)
- The authors assume that in the retrograde setting  the delayed error has access to the current synaptic weights and to $f’(x^\top w)$ at the time of the original activation. This does some heavy lifting for the biological plausibility of the proposed mechanism for delayed learning since it requires that post-synaptic derivatives at the right delayed time, in order for CETs to match them to the right pre-synaptic trace.
- As I understand it, the experiments are a slightly fine-tuned to make the proposed mechanism look good. For example, in the visual experiment the authors consider the time dimension to be the batch dimension and then convolve over it with the CET kernel. This makes the temporal credit-assignment problem much easier than in a genuine online  non-i.i.d. input stream. Later they need salience-based sparsification (top $1.25$% losses) to make the long-delay setting train at all. This sparsification strongly biases what is remembered and is itself a nontrivial extra mechanism, but it’s introduced as an implementation detail rather than a core assumption.

**Questions:**

## Questions

- See weaknesses
- In the retrograde experiments, how is the postsynaptic derivative at $t-T$ stored and retrieved?
- How would you select the optimal CET order and decay parameters?
- In Fig. 5 the authors show that gradient alignment saturates even for high CET orders at long delays. What do you think is the limiting factor and how would you overcome this?
- have you tried the framework on visual experiments with non-i.i.d. input streams?


## Comments


- In the MNIST/CIFAR experiments, the authors state that in the CIFAR dataset the performance of the classical eligibility traces deteriorates at any increasing delay value, while for the MNIST there is some robustness wrt to delays. They conclude that this indicates that “more complex visual tasks are less robust to imprecise time resolution” (lines 256-257). However I am not sure whether one can immediately state this as conclusion. The fact is that the two experiments use widely different architectures (MLP vs CNN)
> We use a 3-layer MLP (input→512 →512 →10) for MNIST and a small CNN with 3 convolutional layers (input →32 →64 →128) and two linear layers (512 →10) for CIFAR-10.

For the MNIST dataset the 512-512 architecture is probably on the generous side of the required size spectrum, and it can probably reach very low error even if the updates are a bit noisy or slightly wrong. The CIFAR model has moderate size for the dataset, and thus needs accurate gradients to get decent accuracy. Corrupting the gradients with temporal smearing results in performance that deteriorates right away with increasing delays in the CIFAR model, but this is not directly related to the fact that the task is complex, but rather because (probably) the model is at its expressivity threshold. While I understand that more complex tasks need larger networks, I would say that to disambiguate whether the drop in performance with increasing delays for the simple eligibility traces is directly related to the complexity of the task or the expressivity of the network for the given task, one should perform the same experiment for increasingly more/less expressive architectures and observe how results change. **I am not asking authors to perform these experiments**, I just mention this as a comment for their argument in lines 256-257. Also I don’t have good intuition on how the difference in the architecture MLP vs CNN could influence the results, but would be grateful if the authors could comment on this.

- The idea of the cascading eligibility traces conceptually reminds me of the much earlier and much more primitive idea of temporal filterbank employed in [1] that shape the effective learning window. I think it would be interesting if the authors could connect their  more conceptually advanced idea to this earlier work.


## Minor


- the authors start several sentences the text with the phase “As well,…” which reads a bit ackward and from my side is the first time I encounter this phrase at the start of a sentence.
- In Figure 3 in vertical axis I would propose to make the axis increasing as you go up in the figure (pot the origin at the  lower left)

---
## References

[1] Porr, Bernd, and Florentin Wörgötter. "Learning with “relevance”: using a third factor to stabilize Hebbian learning." Neural computation 19.10 (2007): 2694-2719.

---

> ### Author Response · Authors · 2025-11-26
>
> Thank you for the review! Here, we summarize the new experimental results, and answer your questions.
>
> > There is no comparison with the eligibility trace formalisms that consider different traces for potentiation and depression (He et al., 2015; Huertas et al., 2016)
>
> > The idea of the cascading eligibility traces conceptually reminds me of the much earlier and much more primitive idea of temporal filterbank employed in [1] that shape the effective learning window. I think it would be interesting if the authors could connect their more conceptually advanced idea to this earlier work.
>
> We appreciate the reviewer’s suggestion to strengthen the connection with prior work on multi-trace models. In response, we have added a theoretical comparison with the two trace LTP-LTD eligibility trace formulation. In Appendix A3 (Proposition 2), we study this kernel as a probability density function and show that **CET(2) is optimal: for a fixed peak, the optimal choice of parameters (a,b) that minimizes the second moment around T converges pointwise and in L1 to the CET(2) kernel**. While it would also be possible to consider the sum of more exponentials and sign-changing kernels, such an extension drastically increases the dynamical system’s complexity in terms of parameters. In contrast, the proposed approach offers a way to scale up the number of states and improve the temporal resolution while keeping a single parameter $\alpha$ that fully specifies the kernel for a fixed $n$.
>
> > The authors assume that in the retrograde setting the delayed error has access to the current synaptic weights and to  at the time of the original activation. This does some heavy lifting for the biological plausibility of the proposed mechanism for delayed learning since it requires that post-synaptic derivatives at the right delayed time, in order for CETs to match them to the right pre-synaptic trace.
>
> > In the retrograde experiments, how is the postsynaptic derivative at  stored and retrieved?
>
> Thank you for your question! One way to approximate this derivative on a slow scale is discussed in (Fan & Mysore, 2024, specifically for retrograde models) and is based on E/I balance. Another option is to use CETs to store the postsynaptic derivative history so that it can be multiplied with upstream error signals propagating backwards. To explore this, we ran two experiments: using another CET (of the same order) to approximate the derivative, and reusing the original CET to estimate it (as this CET stores the pre-post product, it should be small when the derivative is zero), please see Appendix G. In these preliminary experiments, we obtained slightly worse performance compared to the oracle case but CET(10) still solves the environment, and the dependence on CET order behaves as expected: performance decreases for lower orders. See Appendix G for the full results.
>
> Regarding future work, because the derivative signal is shared across all presynaptic neurons, it can be stored in the postsynaptic soma. We assume that the soma would have a much larger capacity than individual synapses (the location where we currently model CETs). Future work could explore performance gains from larger, but not perfect, CET capacity in the soma.
>
> [Fan, Xinhao, and Shreesh P. Mysore. "Quantifying information stored in synaptic connections rather than in firing patterns of neural networks." ArXiv (2024): arXiv-2411.]
>
> > As I understand it, the experiments are a slightly fine-tuned to make the proposed mechanism look good. For example, in the visual experiment the authors consider the time dimension to be the batch dimension and then convolve over it with the CET kernel.This makes the temporal credit-assignment problem much easier than in a genuine online non-i.i.d. input stream.
> > have you tried the framework on visual experiments with non-i.i.d. input streams?
>
> We would like to note that using the time dimension as a batch dimension for the CET kernel in our visual experiments is purely a computational convenience. It does not simplify temporal credit assignment. To evaluate the framework on non-i.i.d. visual streams, we added appendix I with experiments on synthetic videos derived from CIFAR-10 using the BTSP experiment setup. We generate temporally coherent sequences by interpolating intermediate frames and labels and the results show that CETs can perform better with non-i.i.d. inputs. Intuitively, increasing temporal correlation shifts the input spectrum toward lower frequencies, where CETs are particularly effective because they act as low-pass temporal filters whose fidelity improves with $n$. Concretely, we observe that the cosine similarity with the true gradient increases with the number of interpolated frames.

---

> > ### Author Response · Authors · 2025-11-26
> >
> > > Later they need salience-based sparsification (top % losses) to make the long-delay setting train at all. This sparsification strongly biases what is remembered and is itself a nontrivial extra mechanism, but it’s introduced as an implementation detail rather than a core assumption.
> >
> > We appreciate the reviewer’s observation regarding the role of salience based sparsification in our experiments and agree it is an important additional assumption rather than a minor implementation detail. We have therefore changed the wording to make the role of sparsification more explicit (lines 361-369), acknowledging that the sparsification is a core assumption that makes CET work for such a long delay of 2 mins.
> >
> > Additionally we added a paragraph in the discussion highlighting this and explaining that this saliency mechanism reflects a broader class of theories in learning and predictive processing in which a subset of observations, typically those with high prediction error, drive plasticity. This idea is well-established in both neuroscience and machine learning. For example, three-factor learning rules suggest that the third factor can encode surprise, effectively setting learning to zero when there is no surprise (Gerstner et al., 2018).  Large prediction error signal is also associated with greater information gain (Friston 2005; Itti & Baldi 2009; Bogacz 2020; Dabney et al. 2020), while selective consolidation of behaviourally relevant events reduces metabolic cost (van Rossum & Pache 2024).
> >
> > By prioritizing the top-k losses, we operationalize this principle in artificial neural networks. While we do not commit to an exact implementation for the salience signal, the role of salience gating in our model is consistent with theories in which synapses update primarily when either a global neuromodulatory factor signals high behavioural relevance (Bogacz 2020; Dabney et al. 2020), or when dendritic compartments represent  mismatches (Sacramento et al. 2017).
> > In the context of delayed credit assignment, this assumption becomes critical because CETs must store presynaptic traces for minutes. Without prioritising high-salience events, memory buffers become saturated, and gradient estimates deteriorate. Thus, salience-based sparsification is not just computationally convenient but theoretically motivated within normative theories of learning from surprising or behaviourally important events.
> >
> > *References*
> >
> > Gerstner, Wulfram, et al. "Eligibility traces and plasticity on behavioral time scales: experimental support of neohebbian three-factor learning rules." Frontiers in neural circuits 12 (2018)
> >
> > Friston, Karl. "A theory of cortical responses." Philosophical transactions of the Royal Society B: Biological sciences 360.1456 (2005): 815-836.
> >
> > Itti, Laurent, and Pierre Baldi. "Bayesian surprise attracts human attention." Vision research 49.10 (2009): 1295-1306.
> >
> > Bogacz, Rafal. "Dopamine role in learning and action inference." Elife 9 (2020): e53262.
> >
> > Dabney, Will, et al. "A distributional code for value in dopamine-based reinforcement learning." Nature 577.7792 (2020): 671-675.
> >
> > > For the MNIST dataset the 512-512 architecture is probably on the generous side of the required size spectrum, and it can probably reach very low error even if the updates are a bit noisy or slightly wrong...  ...just mention this as a comment for their argument in lines 256-257.
> >
> >
> > We interpret the question as follows: *Larger neural networks have a smoother loss landscape, and loss minimization can be achieved along more directions in parameter space than in smaller networks. This can partly compensate for noisy updates.*
> >
> > We experimented with architectural size (making layers wider or adding more layers) during the early exploration stage for the RL and MNIST experiments, and we did not observe better ET/CET performance when using larger networks. However, we did not perform extensive experiments with larger CNN architectures for CIFAR-10, so we cannot fully disentangle task complexity from network expressivity in that setting.

---

> ### Author Response · Authors · 2025-11-26
>
> > How would you select the optimal CET order and decay parameters?
>
> The decay parameter $\alpha$ is set to be (n - 1) / T to match the delay T (see Section 3, lines 208--210). For a fixed delay, a higher CET order is always better as long as memory and compute are not limited. When memory or computation are limited, the CET order can be chosen empirically for each task: for an easy task with a small delay, an order of 2 should be sufficient, for harder tasks with long delays a higher order may be required to enable any learning.
>
> We also would also like to clarify that our primary goal in this work is not to propose CET as a production-ready algorithm optimized for specific hardware constraints, but rather to test a biologically motivated mechanism for delayed credit assignment in a deep-learning setting. Since we believe it can be implemented in the brain via biochemical cascades (with n representing the number of stages in the cascade; see the Discussion section, the paragraph mentioning ETs and CaMKII), our focus is on its qualitative behavior rather than on deriving an optimal n under particular compute/memory budgets.
>
>
> > In Fig. 5 the authors show that gradient alignment saturates even for high CET orders at long delays. What do you think is the limiting factor and how would you overcome this?
>
> The limiting factor is that even for relatively high orders such as CET(10), the decoding of past activities is still imperfect.
>
> We hypothesize that the solution to this problem is holistic: further development of CETs (e.g., by adding non-linearities to enable more precise decoding), combining them with local losses (which carry less delay due to locality), further development of sparsification, and introducing neural synchronization mechanisms into the network, and so on.

---

### Official Review · Reviewer_SqB5 · 2025-11-05

**Soundness:** 3
**Presentation:** 3
**Contribution:** 3
**Rating:** 6
**Confidence:** 2

**Summary:**

The paper explores state space models as a model for cascading eligibility traces (CETs) to tackle the credit assignment problem of exponentially decaying eligibility traces.
Tuning the decay parameter of their CET lets them decide the time delay at which the CET peaks and assigns maximal credit.

**Strengths:**

- Strong performance with long delays, in some cases even comparable to backpropagation

- CETs always outperform ETs

 - Strong performance on very long delays if input to CETs is zeroed for timesteps with low loss

 - Extensive experiments

- Source code provided for reproducibility

**Weaknesses:**

Since I am not very familiar with the related work literature, I cannot assess the novelty of the work

**Questions:**

Does alpha need to be hand-tuned, or could it be learned if one thinks of a neural network-based application?

---

> ### Author Response · Authors · 2025-11-26
>
> Thank you for the review!
>
> > Does alpha need to be hand-tuned, or could it be learned if one thinks of a neural network-based application?
>
> We thank the reviewer for the question. In Appendix J4, we include experiments demonstrating that the delay parameter $\alpha$ can be learned during training. We consider the case where the delay is fixed but unknown to the network and must be learned. More precisely, we use a biologically plausible weight perturbation rule that approximates gradient descent in expectation. Importantly, Table 16 shows that CETs continue to outperform ETs even when the delay must be learned.

---

### Meta-Review · Area_Chair_FJAD · 2026-01-07

**Summary:**

This paper introduces cascading eligibility traces (CETs) to address the challenge of temporally precise credit assignment. It demonstrates that CETs enable accurate synaptic plasticity models across diverse time scales, from behavioral to extremely slow retrograde signaling.

**Reviewer Concerns:**

The reviewers' common concerns lie in the biological plausibility and implementation details of the proposed mechanism: for example, the need for precise storage/retrieval of postsynaptic information, the reliance on a salience signal to sparsify learning, and the lack of comparison with optimized eligibility trace baselines.

I believe the authors have now provided an effective and comprehensive improvement to the paper. They have addressed the major concerns by adding a discussion on the potential biological basis of the salience signal, including new comparisons with optimized baseline models, and offering more results to improve clarity.

**Reviewer Scores:**

I believe the rebuttal is sufficient to raise the scores. The primary issues have adequately addressed. Therefore, Iwould like to recommend acceptance.

---

### Decision · Program_Chairs · 2026-01-26

Accept (Poster)